# Constraining the response of phytoplankton to zooplankton grazing and photo-acclimation in a temperate shelf sea with a 1-D model - towards S2P3 v8.0

Angela A Bahamondes Dominguez[1], Anna E Hickman[1], Robert Marsh[1], and C Mark Moore[1]

[1]School of Ocean and Earth Sciences, National Oceanography Centre, University of Southampton, European Way, Southampton, SO14 3ZH

**Correspondence:** Angela A Bahamondes Dominguez (aab1g15@soton.ac.uk)

**Abstract.** An established 1-dimensional model of Shelf Sea Physics and Primary Production (S2P3) has been developed into three different new models: S2P3-NPZ which includes a Nutrient-Phytoplankton-Zooplankton (NPZ) framework, where the grazing rate is no longer fixed, but instead varies over time depending on different functions chosen to represent the predator-prey relationship between zooplankton and phytoplankton; S2P3-Photoacclim which includes a representation of the process of photo-acclimation and flexible stoichiometry in phytoplankton; and S2P3 v8.0 which combines the NPZ framework and the variable stoichiometry of phytoplankton at the same time. These model formulations are compared to buoy and CTD observations, as well as zooplankton biomass and *in situ* phytoplankton physiological parameters obtained in the Central Celtic Sea (CCS). Models were calibrated by comparison to observations of the timing and magnitude of the spring phytoplankton bloom, magnitude of the spring zooplankton bloom, and phytoplankton physiological parameters obtained throughout the water column. A sensitivity study was also performed for each model to understand the effects of individual parameters on model dynamics. Results demonstrate that better agreement with biological observations can be obtained through the addition of representations of photo-acclimation, flexible stoichiometry, and grazing provided these can be adequately constrained.

## 1 Introduction

Shelf seas are ocean regions where water depth is less than a few hundred metres ($\sim 200$ m) and represent only $\sim 10\%$ by area of the global ocean. However, these systems have a disproportionate importance because of their exceptionally high biological productivity (Holt and Proctor, 2008), being responsible for 15 to 30% of the total oceanic primary production (PP) (Wollast, 1998; Muller-Karger et al., 2005; Davis et al., 2014). Research vessels and remote, autonomous vehicles have been used to study shelf sea regions such as in the Shelf Sea Biogeochemistry (SSB) research programme (https://www.uk-ssb.org/), whose aim was to increase the understanding of how physical, chemical and biological processes interact on UK and European shelf seas, collecting observations throughout 2014 and 2015 in different regions of the UK shelf sea, although these data are not synoptic (i.e. they are not sampled at different locations simultaneously). To complement the available data from research vessels, ocean models have been used to study and understand marine biogeochemistry, including a variety of high spatial resolution models to represent the biogeochemistry of shelf seas with high complexity and horizontal spatial resolution (Sharples, 1999,

2008; Edwards et al., 2012; Marsh et al., 2015).

Different models have been developed to study plankton communities, ranging from very simple ones, e.g. the Lotka-Volterra competition model (Volterra, 1926; Lotka, 1932) to more sophisticated ones, adding more degrees of complexity by including representation of the physical processes of advection and diffusion, or more complexity in ecosystem functions through representation of different groups of organisms and/or size structure. For example, coupled models such as the Nucleus for European

Modelling of the Ocean (NEMO) and European Regional Seas Ecosystem Model (ERSEM) (Edwards et al., 2012), Regional Oceanic Modelling System (ROMS) (Shchepetkin and McWilliams, 2005), and Finite Volume Community Ocean Model (FV-COM) (Chen et al., 2003). Although complexity can be useful for describing the interacting behaviour of multiple system components, incomplete understanding of the ecology and key processes of the organisms, and the lack of data for validation (Anderson, 2005) can reduce the reliability of predictions. Moreover, simulations of models like NEMO-ERSEM, ROMS, and

FVCOM rely on high-performance computing resources and running multiple sensitivity analyses and experiments is difficult. In contrast, simpler models like the Shelf Sea Physics and Primary Production (S2P3) model (Sharples, 1999; Simpson and Sharples, 2012) have been used to study the dynamics of shelf seas and to simulate seasonal stratification with greater computational efficiency by using a 1-D Nutrient-Phytoplankton (NP) model to represent physical and biogeochemical processes in the water column. In temperate shelf seas, away from advective sources such as the shelf break or plumes from rivers, horizon-

tal processes can be neglected in comparison to vertical processes, thus, although S2P3 does not consider advective fluxes it can make a good representation of the dynamics in the water column for temperate shelf seas (Sharples et al., 2006; Sharples, 2008; Marsh et al., 2015). Understanding and management of shelf sea ecosystems depend on the level of understanding of factors that influence the communities of resident organisms and ecosystem models can provide a useful tool to explore these processes. Computationally inexpensive 1-D models like S2P3 can provide useful tools for investigating how different drivers,

like changes in the physical environment (Sharples et al., 2006), can influence shelf sea biogeochemistry. The simplicity of such models combined with the ability to run multiple experiments and long time series can facilitate wider adoption of such models, including by graduate and undergraduate students (Simpson and Sharples, 2012).

The S2P3 v7.0 model was introduced and developed in the work of Marsh et al. (2015), where it was outlined that further

development of that model would include resolving phytoplankton physiology. In this study, the S2P3 v7.0 model is developed by allowing variations in light intensity to produce phenotypic adjustments in the phytoplankton cells by changing the chlorophyll content of the phytoplankton and, therefore, the cellular absorption cross-section (Macintyre et al., 2002). This phenotypic change in response to variations in the photon flux density is called photo-acclimation (Falkowski and Laroche, 1991; Moore et al., 2006). The main property of photo-acclimation is the reduction of photosynthetic pigment content in response

to increased irradiance (Falkowski and Laroche, 1991). Moreover, changes in nutrient availability can further alter cellular chlorophyll and nitrogen quotas (Droop, 1983; Geider et al., 1998), incorporating a combined representation of these two processes (Geider et al., 1998), this new version of S2P3 v7.0 we term S2P3-Photoacclim and relates phytoplankton growth rates to cell quota (Droop, 1983), describing the light-, nutrient-. and temperature-dependencies of phytoplankton growth rate to

varying ratios of N : C : Chl (Geider et al., 1998). On the other hand, simpler models using an NPZ or NPZD framework, with the use of nutrients, phytoplankton, zooplankton, and detritus as the main model components (e.g., Steele, 1974; Wroblewski et al., 1988; Anderson, 2005) have shown good agreement with observations in terms of chlorophyll and PP, by simulating the timing and magnitude of the spring phytoplankton bloom in different regions of the ocean. Despite their relative simplicity, NPZ models can be a better option to approach an understanding of the physics and biology of an ecosystem, which lead to a further development of the S2P3 v7.0 model where the simplest assumption of a fixed proportion of phytoplankton being grazed and remineralised into the DIN pool (grazing rate) is developed into an NPZ framework (S2P3-NPZ model), using a Holling Type 2 or Ivlev grazing functional response of zooplankton grazing on phytoplankton (Franks, 2002), which shows a saturating response to increasing food.

A combination of photo-acclimation, flexible stoichiometry (S2P3-Photoacclim model), and NPZ framework (S2P3-NPZ model) is then performed to produce a newly developed model called S2P3 v8.0 (Figure 1). This paper presents a thorough analysis in terms of sensitivity to biological parameter values in each new developed model, resulting in differences of the model structure. A comparison between each model demonstrates how structural differences influence the representation of the spring phytoplankton bloom and annual PP. The aim of this paper is to provide a better understanding about the predator-prey relationship between zooplankton and phytoplankton, the effects of photo-acclimation, flexible stoichiometry in a simple 1-D model. Model outputs are compared with observations of phytoplankton responses to physical forcing to illustrate importance of different processes representation.

## 2  Study region and model setup

This study is focused on the Central Celtic Sea (CCS), a region located in the North-Western (NW) European Shelf, which is characterised by its tidally dynamic environment and summer stratification (Pingree et al., 1978; Sharples and Holligan, 2006; Hickman et al., 2012). Daily meteorological data, available from the National Centers for Environmental Predictions (NCEP) Reanalysis data (http://www.esrl.noaa.gov/psd) are used to force the model at the CCS site, located at $49.4°N$, $8.6°W$. Wind speed ($\mathrm{m\,s^{-1}}$), cloud coverage (%), air temperature (°C), and relative humidity (%) variables from this dataset are all used to force each model version.

The following description of model setup is applied to each model structure developed in this work. Tidal components consist of the u-component (semi-major axis) and the v-component (semi-minor axis) for the $M_2$, $S_2$, and $N_2$ tidal constituents. Tidal data are obtained from a fine mesh (12km resolution) covering the UK shelf. Tidal currents are predicted using the Proudman Oceanographic Laboratory Coastal Ocean Modelling Systems (POLCOMS) 3-D shelf model (Holt et al., 2009; Wakelin et al., 2009) with an output extracted for the CCS location. Moreover, each model is initialised on $1^{st}$ January of the first year of simulation with a temperature of 10.10 °C at all depths, and water column presumed mixed throughout, including a vertical

resolution of 1m (i.e. 140 vertical levels). Initial values of physical variables are consistent with former studies (Sharples, 1999, 2008; Marsh et al., 2015), whereas initial values of biological variables are based on observations of zooplankton biomass at the CCS location over winter months of 0.02 mmol N m$^{-3}$ (Giering et al., 2018); phytoplankton chlorophyll correspond to a typical winter value for the CCS location of 0.2 mg Chl m$^{-3}$, and the DIN initial value is 7 mmol N m$^{-3}$. The initialised variables are only set up at the start of each simulation and do not reset in between years.

## 3   Model development

The S2P3 v7.0 model can be divided into two different components: a physical part and a biological part. For this research, the model is an improved version of the original described in Sharples et al. (2006) to be compiled and executed in a Unix environment (Marsh et al., 2015), allowing a 1-D representation of physical and biological processes in shelf seas by simulating the seasonal cycle of phytoplankton, water column stratification, and PP at a selected location defined by water depth and tidal current amplitude. The physical part of the model has been greatly described in many other studies (Sharples, 1999; Sharples et al., 2006; Sharples, 2008; Simpson and Sharples, 2012; Marsh et al., 2015) and is not described in this section again. This model uses the turbulence closure scheme based on Canuto et al. (2001). Likewise, the biological part of the S2P3 v7.0 model is described in Marsh et al. (2015) (Figure 1a).

In order to explicitly account for the influence of zooplankton grazing and, hence, predator-prey dynamics, the S2P3 v7.0 model (Sharples et al., 2006; Marsh et al., 2015) is developed into an NPZ framework. This new version of the model (S2P3-NPZ) includes zooplankton as a state variable, contrary to the S2P3 v7.0 model where grazing ($G$) is calculated as a fixed seasonal cycle represented as a sink term in the phytoplankton tendency equation. Addition of an explicit grazer also allows comparison to zooplankton biomass observations. As within S2P3 v7.0, the biological part of the S2P3-NPZ model calculates phytoplankton biomass in chlorophyll currency ($Phyto_{chl}$; mg Chl m$^{-3}$). Similar to the S2P3 v7.0 model, the growth of phytoplankton biomass ($\mu$) can be either nutrient-limited or temperature-limited, with the maximum growth rate of phytoplankton being related to temperature through an Eppley function and being modified by a nutrient quota ($Q$), which corresponds to a varying ratio between phytoplankton nitrogen and phytoplankton chlorophyll biomass ($Q = Phyto_N/Phyto_{chl}$). Additionally, phytoplankton are modelled in terms of nitrogen (mmol N m$^{-3}$) represented by $Phyto_N$. Zooplankton biomass and external DIN are likewise modelled in terms of nitrogen (mmol N m$^{-3}$; Figure 1b).

The S2P3-Photoacclim model is a new version of the S2P3 v7.0 model, incorporating a more complete representation of phytoplankton physiology (Geider et al., 1998). This model allows phytoplankton to acclimate to changes in light and nutrients, therefore, the ratios of N : C : Chl and characteristics of phytoplankton physiology can vary, allowing direct comparison to physiological data. The biological part of the S2P3-Photoacclim model uses three currencies of phytoplankton biomass: carbon ($Phyto_C$; mg C m$^{-3}$), nitrogen ($Phyto_N$; mg N m$^{-3}$), and chlorophyll ($Phyto_{chl}$; mg Chl m$^{-3}$). The S2P3-Photoacclim

model calculates phytoplankton growth as a function of both nitrogen assimilation and carbon fixation (i.e. variable Chl : N and Chl : C ratios). It is assumed that respiration (R) is equal for all cellular components as a function of temperature: $R_C = R_n = R_{chl} = R_{ref}T_{function}$, where $R_{ref}$ (d$^{-1}$) is a degradation rate constant at a reference temperature (Figure 1c).

The S2P3 v8.0 model describes a combination of zooplankton and physiological acclimation components in order to provide a more realistic representation of the ecosystem dynamics (Figure 1d). This model can be divided into two different components: a physical part and a biological part. Full details and model equations of the biological part of the model are described below. Details of the variables and parameters are listed in Appendix A.

The biological part of the S2P3 v8.0 model calculates changes in phytoplankton carbon biomass ($Phyto_C$) over time as:

$$\frac{\partial Phyto_C}{\partial t} = \frac{\partial}{\partial z}\left(K_Z \frac{\partial Phyto_C}{\partial z}\right) + Phyto_C(\mu - R_C T_{func} - u\zeta) - I\frac{Z}{Q_P}, \tag{1}$$

where $\mu$ is the growth rate of phytoplankton, $R_C$ is a respiration rate constant of phytoplankton, $T_{func}$ is a temperature-response function of phytoplankton, $u$ is the phytoplankton carbon-specific nitrate uptake rate, $\zeta$ is the cost of biosynthesis, $I$ is the ingestion rate of zooplankton, and $Q_P$ is the cellular nutrient : carbon quota.

Phytoplankton biomass is also modelled in terms of internal nitrogen, $Phyto_N$. S2P3 v8.0 calculates $Phyto_N$ as:

$$\frac{\partial Phyto_N}{\partial t} = \frac{\partial}{\partial z}\left(K_Z \frac{\partial Phyto_N}{\partial z}\right) + uPhyto_C - Phyto_N(R_n T_{func}) - IZ, \tag{2}$$

where $R_n$ is the nitrate remineralisation constant.

The rate of change of phytoplankton biomass in terms of chlorophyll ($Phyto_{chl}$) is described by:

$$\frac{\partial Phyto_{chl}}{\partial t} = \frac{\partial}{\partial z}\left(K_Z \frac{\partial Phyto_{chl}}{\partial z}\right) + u\rho_{chl}Phyto_C - Phyto_{chl}(R_{chl}T_{func}) - I\frac{Z}{Q}, \tag{3}$$

where $\rho_{chl}$ is a chlorophyll synthesis regulation term, $R_{chl}$ is the chlorophyll degradation rate constant, and $Q$ is the cellular nutrient quota (N : Chl).

The change in time of external DIN is calculated as:

$$\frac{\partial N}{\partial t} = \frac{\partial}{\partial z}\left(K_Z \frac{\partial N}{\partial z}\right) + \gamma_1 IZ + \gamma_2 mZ + Phyto_N(R_n T_{func}) - uPhyto_C, \tag{4}$$

where $\gamma_1$ is the grazing inefficiency or 'messy feeding' that returns a fraction of grazed material back into the DIN pool, $\gamma_2$ is the fraction of dead zooplankton that goes into the sediments, and $m$ is the loss rate of zooplankton due to predation and

physiological death.

Zooplankton grazing on phytoplankton response depends on a Holling Type II or Ivlev grazing (Franks, 2002), with the ingestion rate of zooplankton ($I$) described as:

$$I = R_m(1 - e^{(-\lambda Phyto_N)}),$$ (5)

where $R_m$ is the zooplankton maximal grazing rate ($\text{d}^{-1}$) and $\lambda$ is the rate at which saturation is achieved with increasing food levels (mmol N $\text{m}^{-3}$)$^{-1}$.

Zooplankton biomass is, therefore, modelled as:

$$\frac{\partial Z}{\partial t} = \frac{\partial}{\partial z}\left(K_Z \frac{\partial Z}{\partial z}\right) + (1 - \gamma_1)IZ - mZ.$$ (6)

The carbon-specific, light-saturated photosynthetic rate depends on the internal nitrogen of phytoplankton described as a factor f based on the work of Moore et al. (2001):

$$P_m = P_{max}^C f,$$ (7)

where $P_{max}^C$ is the maximum value of the carbon-specific rate of photosynthesis and the factor f is defined as:

$$f = \frac{Q - Q_{min}}{Q_m - Q_{min}}.$$ (8)

Nitrogen assimilation is calculated as a saturating function of external nutrient content, the internal nutrient quota and the maximum nitrogen assimilation rate ($u_m$) according to (Geider et al., 1998; Moore et al., 2001):

$$u = u_m\left(\frac{1 - f}{1.015 - f}\right)\left(\frac{N}{k_n + N}\right),$$ (9)

where $k_n$ is the half-saturation constant for nitrate uptake.

The carbon-specific photosynthesis is a saturating function of irradiance and it is calculated as:

$$\mu = P_m(1 - e^{-\left(\frac{\alpha^{chl}I_{PAR}\theta}{P_m}\right)}),$$ (10)

where $\alpha^{chl}$ is the chlorophyll-specific initial slope of the photosynthesis-light curve, $\theta$ is the cellular chlorophyll : phytoplankton carbon ratio, and $I_{PAR}$ is the photosynthetically available radiation.

Finally, chlorophyll-*a* synthesis depends on the rates of photosynthesis and light absorption:

$$\rho_{chl} = \theta^N_{max} \left( \frac{\mu}{\alpha^{chl} I_{PAR} \theta} \right), \tag{11}$$

where $\theta^N_{max}$ is the maximum value of the cellular chlorophyll : phytoplankton nitrogen ratio.

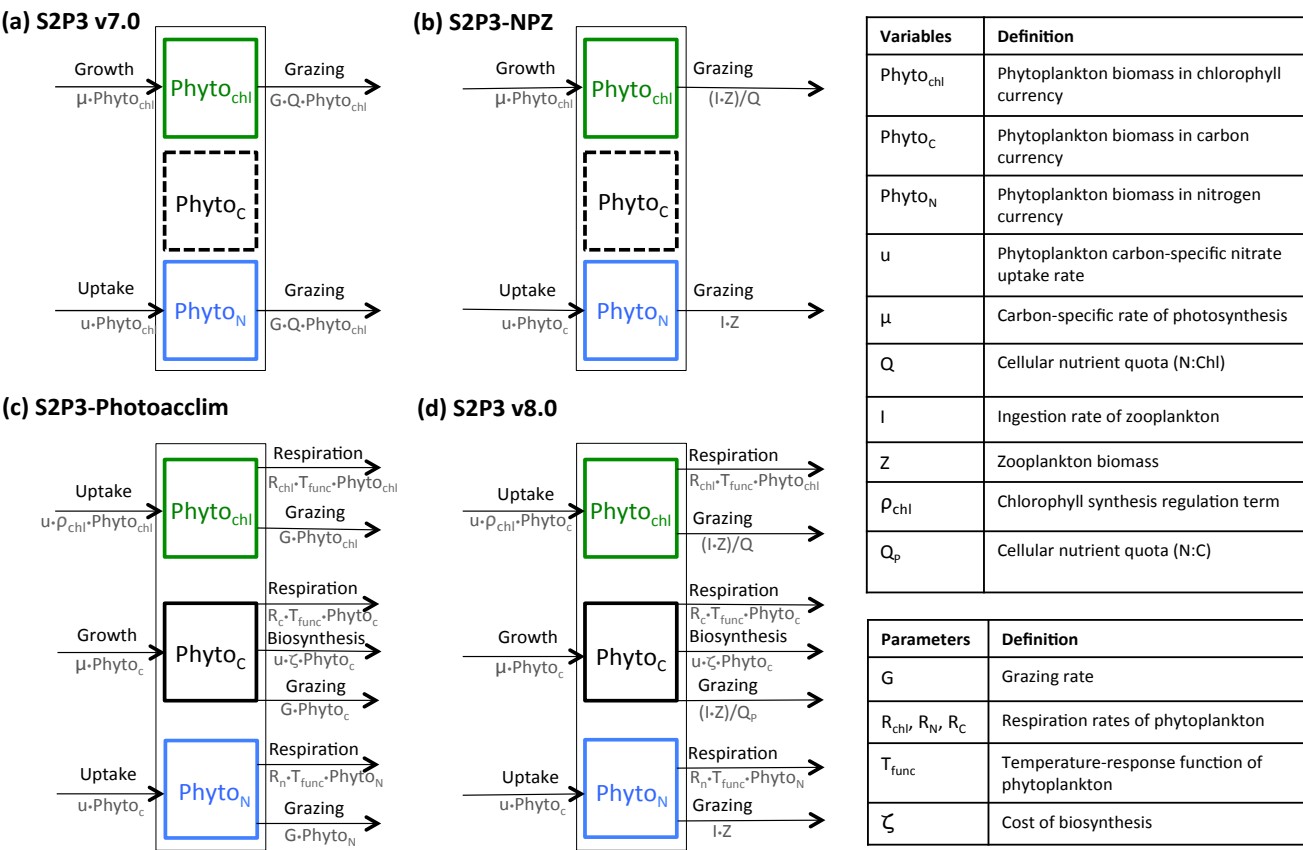

**Figure 1.** Structure of the phytoplankton growth formulations: (a) S2P3 v7.0 model, with constant $Phyto_N : Phyto_{chl}$ ratio (Q); (b) S2P3-NPZ model, including explicit zooplankton with an associated ingestion rate of phytoplankton (I) based on the Ivlev grazing type; (c) S2P3-Photoacclim model, with varying ratios of N : C : Chl, with phytoplankton chlorophyll content regulated by a coefficient of chlorophyll synthesis ($\rho_{chl}$), which reflects the ratio of energy assimilated to energy absorbed (Geider et al., 1996), and with phytoplankton carbon regulated through a cost associated with biosynthesis ($u\zeta$) (Vries et al., 1974; Geider, 1992; Geider et al., 1998), respiration, and grazing (G); (d) S2P3 v8.0 model, including two different varying quotas for $Phyto_N : Phyto_c$ and $Phyto_N : Phyto_{chl}$.

## 4    Validation of the models: observations

To calibrate or tune each model, they were adjusted on a trial-and-error basis until disagreement with the *in situ* observations was minimised, allowing investigation of the sensitivity of each model to changes in the parameters listed in Table 1.

## 4.1 UK SSB programme

Time-series of surface chlorophyll-*a* concentrations (mg Chl m$^{-3}$) from long-term mooring deployments including the Carbon and Nutrient Dynamics and Fluxes over Shelf Systems (CaNDyFloSS) Smartbuoy (Mills et al., 2003) were collected at the CCS location (49.4°N, 8.6°W, depth 145.8 m), gathering data for 5 minutes every 30 minutes during the years 2014 and 2015 as part of the research cruise expeditions DY029 and DY033. The phytoplankton community fluorescence from the water samples were calculated as a proxy for chlorophyll-*a* and calibrated taking into account daytime fluorescence quenching which results in a reduction of fluorescence per unit chlorophyll. For this study, day time data was removed.

CTD casts were performed in different locations of the NW European shelf from the CCS location to the shelf break, with discrete samples of temperature, DIN, and chlorophyll-*a* collected using Niskin bottles as part of the research cruise expeditions DY029 and DY033. At the CCS location, the CTD samples were collected from pre-dawn to midday with a 1m vertical resolution over the whole water column (140m depth) during the year 2015. CTD casts for the CCS location were chosen to validate the model during spring and summer. Relevant information about dates and positions from other CTD casts taken during spring and summer of the year 2015 are listed in Table A1. CTD casts chosen to compare the each model are marked in bold in Table B1.

## 4.2 Zooplankton biomass

Zooplankton biomass samples were collected at the CCS location (49°25 N, 8°35 W, $\sim 150$ m water depth) during four periods: 5th - 12th August 2014, 10th - 29th November 2014, 3rd - 28th April 2015, and 13th - 31st July 2015 for the cruises DY026, DY018, DY029, and DY033, respectively (Giering et al., 2018). Zooplankton were fractionated into microplankton, small mesozooplankton, and large mesozooplankton by using different mesh sizes. For zooplankton biomass samples, net rings of 57 cm diameter were used and fitted with two different mesh sizes of 63 $\mu$m and 200 $\mu$m. The nets had a closing mechanism when deployed, sampling zooplankton biomass during daytime and night-time at different depth: above and below the thermocline, and, when present, across the deep chlorophyll maximum (DCM; determined based on fluorescence measurements). The thermocline and DCM were determined from CTD casts immediately prior to the net deployments. The 63 $\mu$m and 200 $\mu$m mesh nets were hauled at 0.2 m s$^{-1}$ and 0.5 m s$^{-1}$, respectively.

For the S2P3-NPZ and the S2P3 v8.0 models, only mesozooplankton biomass were considered, with a community composition that included: amphipods, appendicularian, chaetogratha, copepods, euphausiacea, polychaeta, and others (e.g. cladocerans, dinoflagellates, echinoderm, eggs, foraminifera, gymnosomata, unidentified larvae, nauplii, ostracods and radiolarian, all of which contributed $< 3\%$ in all samples). A FlowCam (Fluid Imaging Technologies Inc.) and a ZooScan were used to scan zooplankton individuals with images processed using ZooProcess 7.19 and Plankton Identifier 1.3.4 softwares (Gorsky et al., 2010). From these images, biovolume spectra were calculated and converted into image-derived dry weight (DW). The total

246 net hauls collected for biomass samples provided 44 vertical depth profiles, integrating zooplankton biomass typically
between 0 and 120 m at the CCS location. The complete data set can be obtained from the British Oceanographic Data Centre
(BODC), (http://www.bodc.ac.uk/data) as reported in Giering et al. (2018).

### 4.3   Physiological observations

Water samples were collected in the Celtic Sea from the cruises JR98 and CD173 using 10-liter Niskin bottles for four of the
sites: CS1, CS2, CS3, and IS1 (Figure 2) during 24-h periods on 31$^{st}$ July, 29$^{th}$ July, 05$^{th}$ August, and 2$^{nd}$ August, respec-
tively. Multiple samples were collected in the surface and in deeper layers (in the surface mixed layer (SML) and the DCM)
to obtain different phytoplankton populations throughout the photoperiod. The JR98 cruise was undertaken from 24$^{th}$ July to
14$^{th}$ August of 2003. Stations, from the Irish Sea to the Celtic Sea shelf break, ranged in characteristics from very strong,
narrow thermoclines in the southern Celtic Sea (CS1), to the weak, deep surface layer associated with internal wave mixing
at the shelf edge (CS2). On the other hand, the CD173 cruise was undertaken from 15$^{th}$ July to 6$^{th}$ August of 2005, from
the stratified region of the Celtic Sea shelf (stations D2, CS1, CS3, U2) and shelf break (stations CS2, N1). For this work,
observations from both cruises were used, considering only the stations from the seasonally stratified sites (B2, CS1, CS3, D2,
JB1, OB, P1, U2, and ctd16) and excluding stations CS2 and N1 which are close to the shelf edge where advective fluxes are
more relevant than in the stations nearer to the CCS location and none of the models used here consider advective fluxes.

Photosynthesis versus irradiance (P vs E) experiments were conducted in short-term incubations (2 - 4h) using a photosyn-
thetron (Moore et al., 2006). From these P vs E experiments chlorophyll-$a$ normalised PP was derived from $^{14}$C uptake to obtain
the chlorophyll-$a$ specific maximum light-saturated photosynthesis rate $P_{max}^{Chl}$ (mg C (mg Chl $-$ a)$^{-1}$ h$^{-1}$) and the maximum
light utilisation coefficient, $\alpha^{chl}$ (mg C (mg Chl-a)$^{-1}$ h$^{-1}$ ($\mu$E m$^{-2}$ s$^{-1}$)$^{-1}$) (Jassby and Platt, 1976; Hickman et al., 2012).
Values of $\alpha^{chl}$ and the light saturation parameter, $E_k$ ($\mu$E m$^{-2}$ s$^{-1}$) (given by $E_k = P_{max}^{Chl}/\alpha^{chl}$) were spectrally corrected to
the *in situ* irradiance at the sample depth according to the phytoplankton light absorption (Moore et al., 2006). The maximum
light utilisation coefficient ($\alpha^{chl}$) was constrained for the S2P3-Photoacclim and the S2P3 v8.0 models by finding the mean
from all observations: $\alpha^{chl} = 9.16 \times 10^{-6}$ mg C (mg Chl-a)$^{-1}$ h$^{-1}$ ($\mu$E m$^{-2}$ s$^{-1}$)$^{-1}$ (ranges of this parameter accounted from
$3.58 \times 10^{-6}$ to $3.59 \times 10^{-5}$ mg C (mg Chl-a)$^{-1}$ h$^{-1}$ ($\mu$E m$^{-2}$ s$^{-1}$)$^{-1}$; std $= 4.38 \times 10^{-6}$), and therefore, the values of $P_{max}^{Chl}$
and $E_k$ were used as variables for comparison with equivalent modelled values.

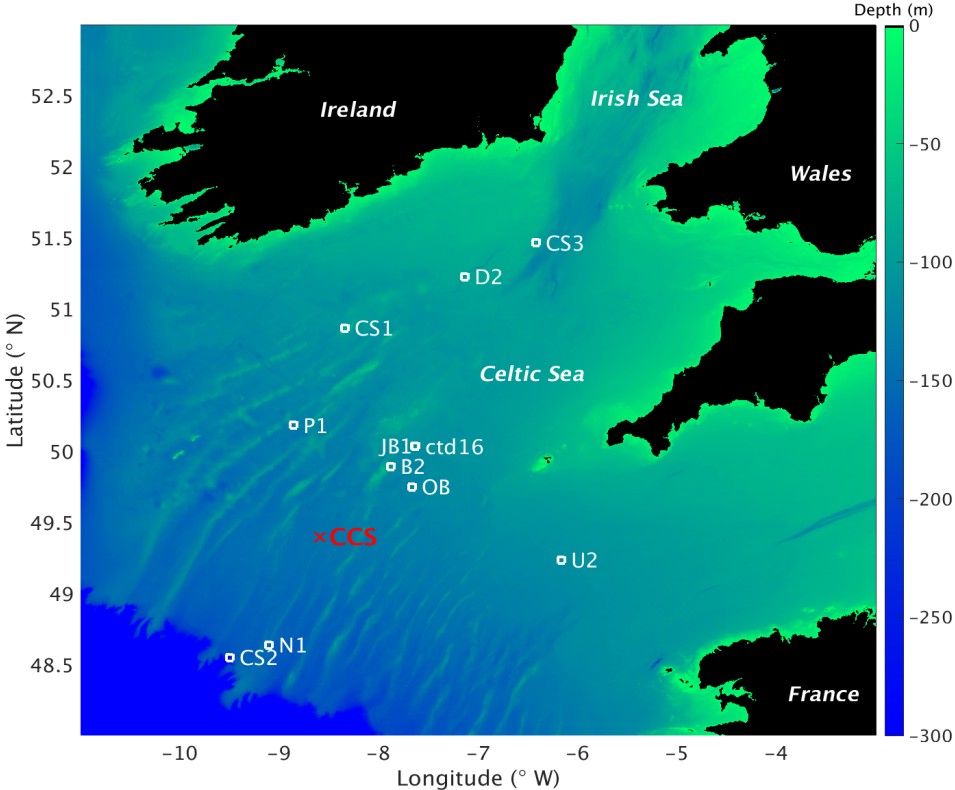

**Figure 2.** Map for study area and stations for the JR98 and CD173 cruises, including the CCS location (in red colour). Image created with Matlab using the repository data for gridded bathymetry provided by General Bathymetric Chart of the Oceans (GEBCO). Bathymetric data only considered for the shelf sea region (0 to 300m depth) with open ocean depth neglected (deeper than 300m). Continents considered in black colour (over 0m elevation).

| Parameters (Units) | Definition | S2P3-NPZ | S2P3-Photoacclim | S2P3 v8.0 |
|---|---|---|---|---|
| $\gamma_1$ (dimensionless) | Grazing inefficiency or 'messy feeding' (0.0-1.0), returns a fraction of grazed material back into the DIN pool | 0.2 | | 0.1 |
| $\gamma_2$ (dimensionless) | Fraction of dead zooplankton (0.0-1.0) that goes into the sediments or higher trophic levels | 0.5 | | 0.4 |
| $\lambda$ (mmol N m$^{-3}$)$^{-1}$ | Rate at which saturation is achieved with increasing food levels | 0.053 | | 0.014 |
| $R_m$ (d$^{-1}$) | Zooplankton maximal grazing rate | 2.5 | | 3.5 |
| m (d$^{-1}$) | Loss rate of zooplankton due to predation and physiological death | 0.05 | | 0.02 |
| $P_{max}^C$ (d$^{-1}$) | Maximum value of the carbon-specific rate of photosynthesis | | 2.0 | 3.5 |
| $Q_m$ (mmol N (mg C)$^{-1}$) | Maximum value of the cellular nutrient quota | | 0.028 | 0.032 |
| $\theta_{max}^N$ (mg Chl (mmol N)$^{-1}$) | Maximum value of the chlorophyll : phytoplankton nitrogen ratio | | 4.2 | 2.1 |
| $R_C = R_n = R_{chl}$ (d$^{-1}$) | Respiration rates | | 0.02 | 0.02 |

**Table 1.** List of parameter values, including units and definitions for the calibrated S2P3-NPZ, S2P3-Photoacclim, and S2P3 v8.0 models.

## 4.4   Model calibration

For this study approximately two years of phytoplankton chlorophyll data were available for the CCS location, while in the case of the zooplankton biomass observations, these were collected only during certain days per year allowing only a discrete representation of the seasonal cycle of zooplankton. Finally, profiles of physiological data were only collected during summertime of the years 2003 and 2005. The calibrated version of the S2P3-NPZ model shows differences in the timing of the spring phytoplankton bloom for the year 2015 in comparison to observations of surface chlorophyll (Figure 3a), with a later bloom from the S2P3-NPZ model, reaching a peak bloom about a month later. Additionally, the magnitude of the spring phytoplankton bloom is also higher in the model in comparison to observations. Phytoplankton are able to escape grazing control in April and early May, with the spring zooplankton bloom occurring about a month later (Figure 3b). Similar differences can be observed between the calibrated S2P3-Photoacclim model, demonstrating that constraining the timing of the spring phytoplankton bloom is a complex process. Tuning model parameters to generate earlier blooms modifies the magnitude of the spring phytoplankton bloom by increasing it to unrealistic levels. Figure 3 shows that between the three models the best

agreement found in comparison to buoy observations correspond to the S2P3 v8.0 model, although the timing and magnitude of the spring phytoplankton bloom show some remaining small differences, with higher concentrations of surface chlorophyll-*a* during spring ($\sim 10$ mg Chl m$^{-3}$). Moreover, the timing of the spring phytoplankton bloom during the year 2014 matches the observations but a delayed bloom is shown during the year 2015. On the other hand, zooplankton biomass is higher than in the S2P3-NPZ and the predator-prey relationship is well represented, with the spring zooplankton bloom happening approximately a month later than the spring phytoplankton bloom during the year 2014, but this difference is less during the year 2015, with the start of the zooplankton bloom happening about half a month after the start of the spring phytoplankton bloom. Remaining mismatches between the calibrated models and observations may be driven by water column processes including advection and diffusion that were not considered in these 1-D models, but affects the real water column where the observations were taken. Furthermore, photo-acclimation and grazing depend on changes in temperature (Geider, 1987; Vázquez-Domínguez et al., 2013) and other ecosystem processes which are not explicitly represented, potentially explaining remaining mismatches between the observations and the calibrated S2P3 v8.0 model.

To provide a quantitative index of bloom timing we consider a threshold criteria (Siegel et al., 2002; Greve et al., 2005; Fleming and Kaitala, 2006; Henson et al., 2009). In this study, the spring phytoplankton bloom is defined as when surface chlorophyll reaches more than 1.5 mg Chl m$^{-3}$ (see Table B2). For the S2P3-Photoacclim model in comparison to CTD cast during spring (Figure 4b), the model has not yet reached the spring phytoplankton bloom whereas observations indicate the spring phytoplankton bloom has already started; the S2P3-Photoacclim model shows low chlorophyll-*a* concentrations at the surface, with DIN concentrations not being depleted at this stage (Figure 4c). Similar results can be seen for the S2P3-NPZ, although Table B2 shows that the spring phytoplankton bloom has already started in this model, but it is still later than in the observations. The S2P3 v8.0 model, on the contrary, shows a better agreement in terms of the timing of the spring phytoplankton bloom with the CTD observations. During summer months, the models are able to reproduce the sub-surface mixed layer observed in the CTD profile (Figure 4e), with a similar magnitude but shallower by approximately 20 m. On the other hand, the physical structure of the model shows a good agreement with the observations during spring (Figure 4a), but there are differences during summer (Figure 4d). The lack of a marked mixed layer depth in all the models is likely related to short term meteorology forcing: during the 24/07/2015, the air temperature was high ($\sim 20\,°C$) and wind speed was low ($\sim 5$ m $s^{-1}$). However, the thermocline shows a sharp development in the CTD observations that can not be constrained better in the models by parameterising values of the turbulent closure scheme, light attenuation in the water column, and mixing control parameters (data not shown).

The S2P3-Photoacclim and the S2P3 v8.0 models were further compared with phytoplankton physiological variability observations (Figure 5). Figure 5 shows that near the sea surface, where light levels are high, photo-acclimation of phytoplankton, values of $P_{max}^{Chl}$ and $E_k$ are higher than in deeper layers of the water column. The observed $P_{max}^{Chl}$ ranges approximately 0.5-2.5 $\times 10^{-3}$ (mg C (mg Chl $-$ a)$^{-1}$ s$^{-1}$) in the surface waters (first 5m), while this range is smaller in deeper layers (0-1.0 $\times 10^{-3}$ mg C (mg Chl$-$a)$^{-1}$ s$^{-1}$), e.g. at 40m depth. Similar variability can be observed with $E_k$, having lower values in deeper layers of the water column, but the largest variability occurs in the surface layer ($\sim$ 100 - 250 $\mu$E m$^{-2}$ s$^{-1}$). The S2P3-Photoacclim

and S2P3 v8.0 models show a good agreement with the observations, with plausible values of $P_{\max}^{chl}$ and $E_k$ found through the water column particularly in terms of the magnitude of the vertical gradients. However, the version of the S2P3 v8.0 model which has the best parameterisations when tuned to fit the timing of the spring bloom has remaining discrepancies when compared to the physiological observations, showing an overestimation of $P_{\max}^{Chl}$ and $E_k$ at depth. Calibration of these models against physiological data are relatively novel as such comparisons remain rare. Greater complexity allowed the S2P3 v8.0 model to resolve a more diverse range of biogeochemical dynamics, explicitly accounting for zooplankton biomass and for the dynamics of internal quotas of phytoplanktonic cells, with phytoplankton biomass being in carbon, nitrogen, and chlorophyll currencies, allowing the decoupling of nutrient uptake from carbon fixation (Klausmeier et al., 2004; Flynn, 2008; Bougaran et al., 2010; Bernard, 2011; Mairet et al., 2011; Ayata et al., 2013). Including additional parameters in models can add more unconstrained degrees of freedom (Ward et al., 2010), but also allows for more parameter combinations and, therefore, more flexibility to constrain S2P3 v8.0 in order to reproduce observations. The new added parameters and variables in the S2P3 v8.0 model, were carefully chosen to allow the model to be constrained against additional data, specifically zooplankton abundances and photosynthetic physiological measurements, therefore, a higher complexity allowed better representations of the temporal dynamics. Additionally, despite having more sophisticated formulations of the ecosystem, the S2P3 v8.0 model continues to be a 1-D model, allowing multiple experiments to be run at the same time with relatively low computational cost.

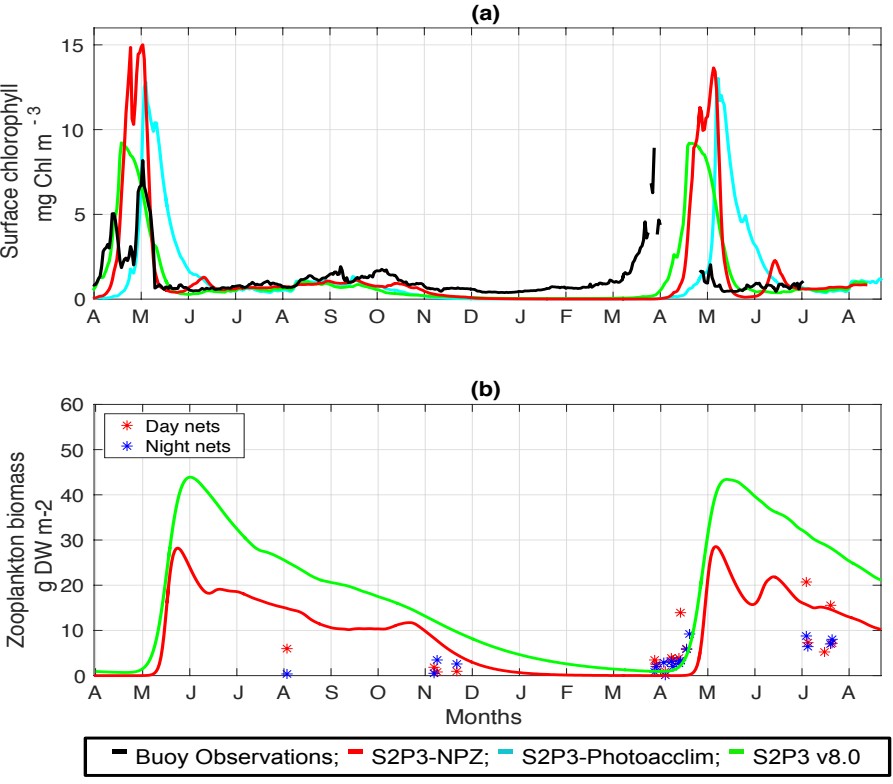

**Figure 3.** (a) SSB observations of surface chlorophyll-*a* (black line), along with the modelled surface chlorophyll-*a* for the S2P3-NPZ (red line), S2P3-Photoacclim (cyan line), and S2P3 v8.0 (green line) calibrated models. (b) Observations of zooplankton biomass presented as discrete points for day nets (red dots) and night nets (blue dots) taken during the cruises DY026, DY018, DY029, and DY033; modelled zooplankton biomass from the S2P3-NPZ (red line) and S2P3 v8.0 (green line) calibrated models.

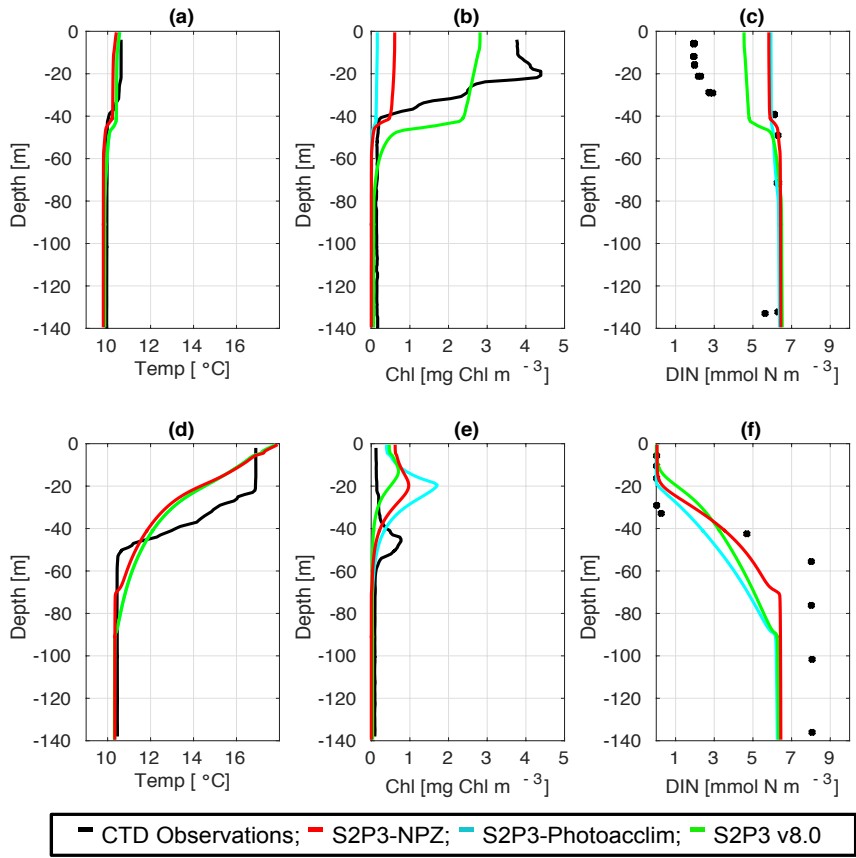

**Figure 4.** CTD observations from the SSB programme (black lines) including data for: springtime (20/04/2015) (a) temperature, (b) chlorophyll-*a*, and (c) DIN (black dots); for summertime (24/07/2015) (d) temperature, (e) chlorophyll-*a*, and (f) DIN (black dots) along the S2P3-NPZ (red line), S2P3-Photoacclim (cyan line), and S2P3 v8.0 (green line) calibrated models.

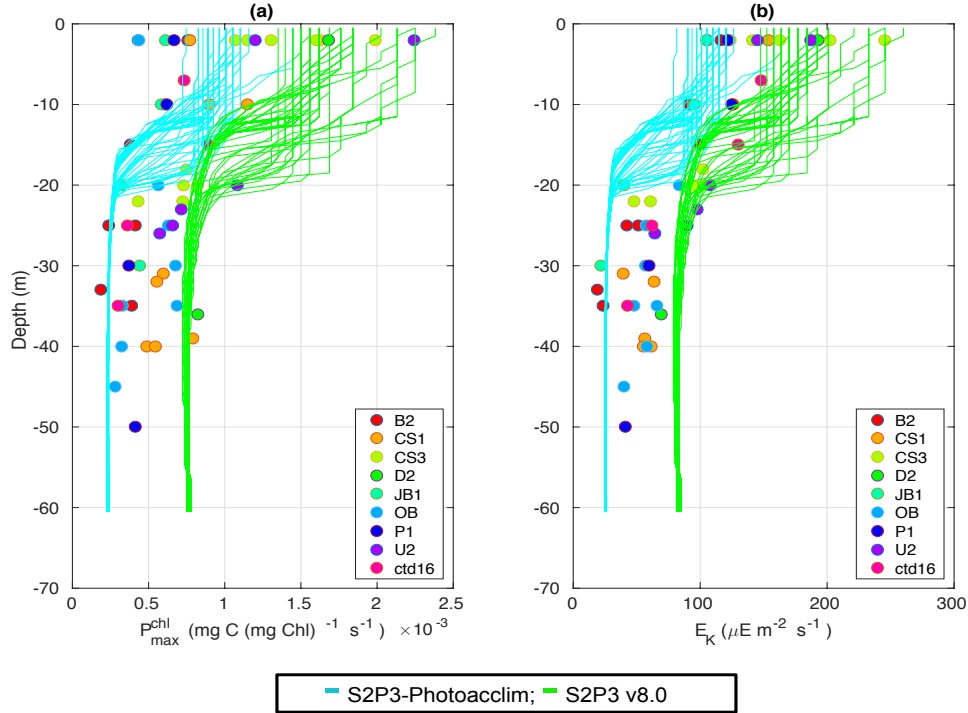

**Figure 5.** Observations from the cruises CD173 and JR98 for: (a) chlorophyll-a specific maximum light-saturated photosynthesis rate ($P_{max}^{Chl}$) in different locations of the Celtic Sea and for the calibrated S2P3-Photoacclim model (cyan lines) and the S2P3 v8.0 model (green lines); (b) observations of the light saturation parameter ($E_k$) for different stations across the Celtic Sea and for the calibrated S2P3-Photoacclim model (cyan lines) and the S2P3 v8.0 model (green lines). The data from both models were plotted for the same days that the observations were collected.

The behaviour of the S2P3 v8.0 model calibrated for the CCS location is displayed in Figures 6 and 7. Figure 6 shows contour plots from daily profiles of the S2P3 v8.0 model for temperature (Figure 6a), phytoplankton chlorophyll-a (Figure 6b), zooplankton biomass (Figure 6c), phytoplankton chlorophyll : phytoplankton carbon ratio (Figure 6d), and DIN (Figure 6e) for the years 2014 and 2015 which correspond to the observation period of the SSB programme. Figure 6 allows a more detailed overview of the model dynamics. Water column temperature increases from April of each year, reaching a maximum value at the surface during summer months. At the same time, the spring phytoplankton bloom can be observed during April, reaching $\sim 10$ mg Chl m$^{-3}$. Additionally, the spring zooplankton bloom can be observed approximately a month after the spring phytoplankton bloom is developed, with zooplankton being able to grow during summer months and decreasing until a minimum value during winter. These spring blooms also mark the start of DIN depletion at the surface, a state that lasts until the end of summer. Finally, the phytoplankton chlorophyll : phytoplankton carbon ratio shows the highest values during winter months when irradiance levels are low, with the Chl : N ratio decreasing until the end of summer due to the lower concentrations of

chlorophyll in the cell to avoid internal damage due to high irradiance during this period, highlighting the photo-acclimation of phytoplankton and flexible stoichiometry of the S2P3 v8.0 model.

Figure 7 provides a general overview of the dynamics of the calibrated S2P3 v8.0 model, representing the inter-annual variability of each variable using the median (black lines), and lower and upper quantiles (red lines; 95% of the data distribution) during 1965 to 2015. Figure 7a shows the start of thermal stratification during early spring with observable inter-annual variability in the extent of stratification. Once the water column is stratified, the spring phytoplankton bloom can be observed (Figure 7b), followed by the start of the spring zooplankton bloom (Figure 7c). As phytoplankton grows, DIN concentrations at the surface start to deplete reaching a minimum value during spring and summer months, and increasing when thermal stratification breaks down during winter months (Figure 7d). Finally, net primary production (NPP) time-series show seasonal and inter-annual phytoplankton dynamics (Figure 7e). All variables of the S2P3 v8.0 model shown in Figure 7 provide the spectrum of inter-annual variability of 95% of the data using the upper and lower quantile values (red lines), demonstrating that the inter-annual variability of thermal stratification provides variability for the timing and magnitude of the spring phytoplankton bloom as well as summer growth.

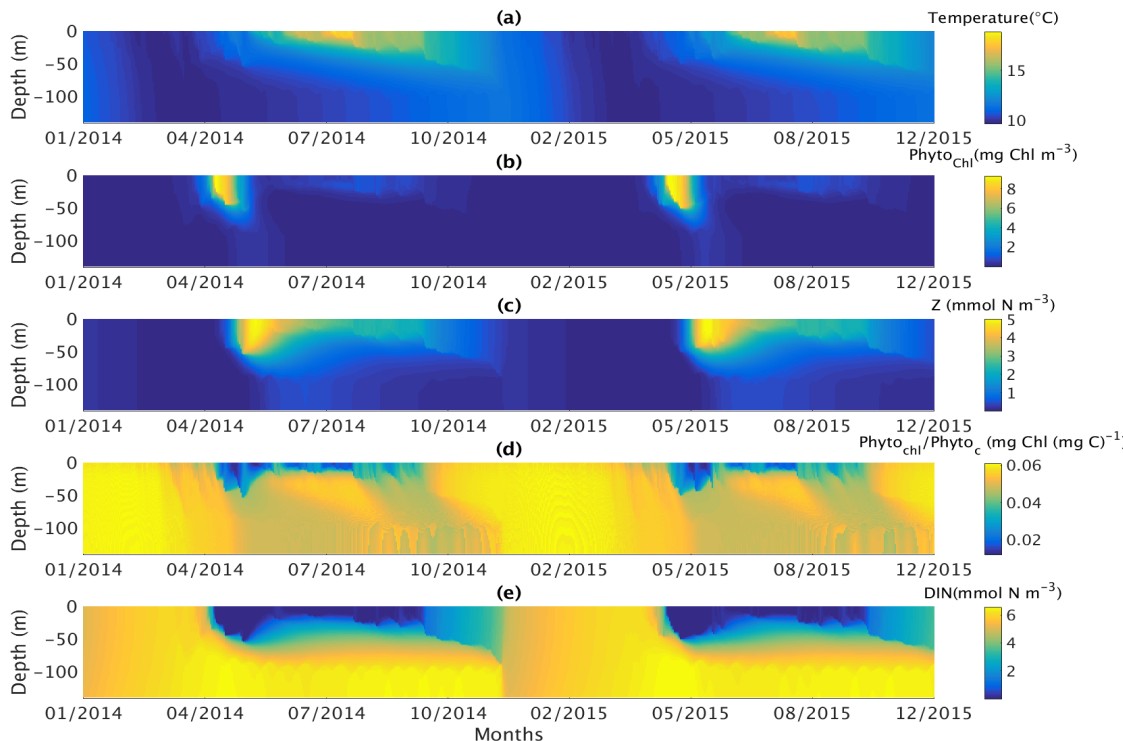

**Figure 6.** Contoured daily vertical profiles for the start of 2014 to the end of 2015 for the calibrated version of the S2P3 v8.0 model including: (a) temperature ($^{\circ}$C), (b) phytoplankton chlorophyll-a (mg Chl m$^{-3}$), (c) zooplankton biomass (mmol N m$^{-3}$), (d) phytoplankton chlorophyll : phytoplankton carbon ratio (mg Chl (mg C)$^{-1}$), and (e) DIN (mmol N m$^{-3}$).

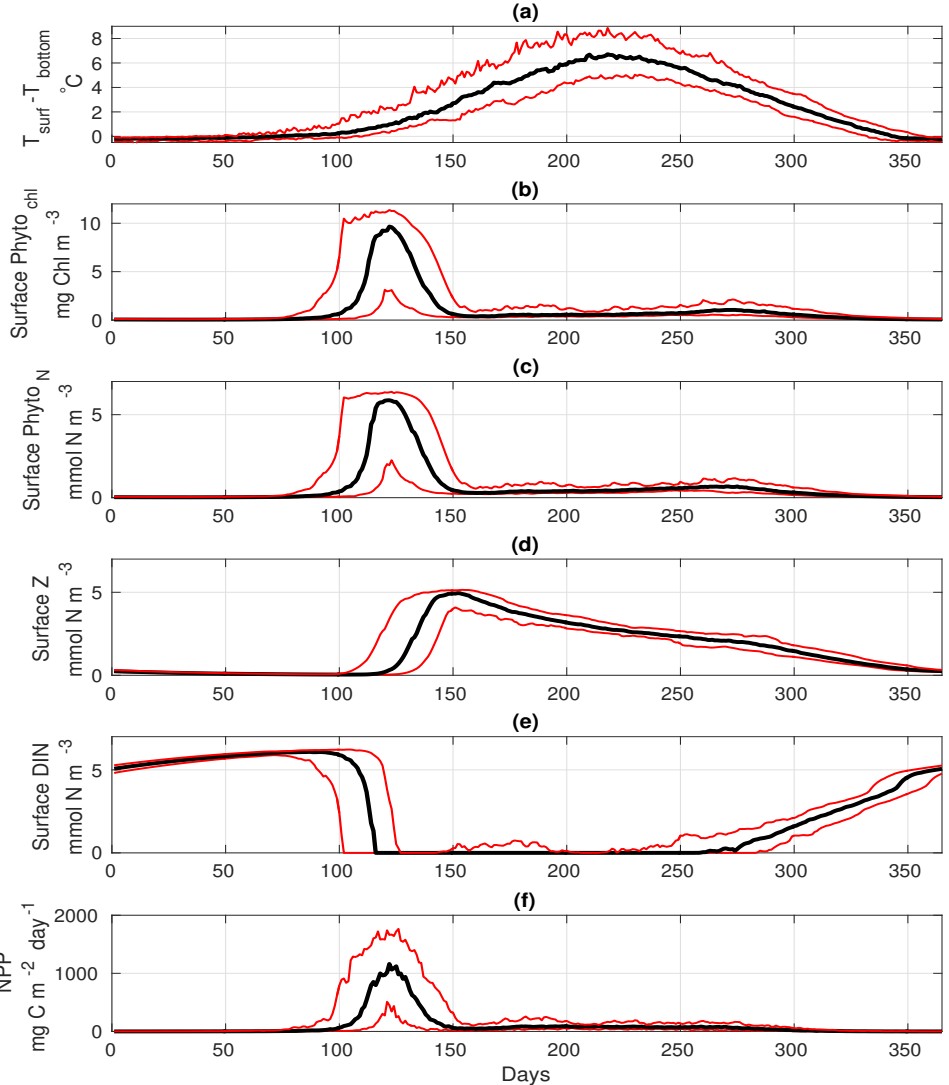

**Figure 7.** Annual representation of the median (black lines) calculated from 1965-2015 for the S2P3 v8.0 model calibrated for the CCS location and forced with all meteorological components (i.e. wind speed, cloud coverage, air temperature, and relative humidity). Red lines represent the annual lower and upper quantiles for each variable of the model (95% distribution of the data) over 1965-2015. (a) Surface temperature minus bottom temperature ($^{\circ}$C), (b) surface chlorophyll-a (mg Chl m$^{-3}$), (c) surface zooplankton biomass (mmol N m$^{-3}$), (d) surface DIN (mmol N m$^{-3}$), and (e) net primary production (NPP) (mg C m$^{-2}$ day$^{-1}$).

## 5 Sensitivity studies

An analysis was performed to assess the sensitivity of the model to selected parameter values for the S2P3 v8.0 model. Each parameter listed in Table 1 was varied in turn in order to understand how sensitive the model is to those changes and the effect that they have on the modelled ecosystem dynamics at the CCS location. In each case parameters were varied from the best calibrated value by +50% and -50%. Sensitivity studies are important tools to improve the accuracy of shelf sea models (Chen et al., 2013), but developing these analyses has to be done carefully in order to identify which processes are responsible for the observed model behaviour (Ward et al., 2013). A direct comparison was calculated in terms of the S2P3 v8.0 model attributes considering: the timing and magnitude of the spring phytoplankton bloom, and the total annual zooplankton biomass for the calibrated version of the model and each experiment, providing better insights on the effects that each parameter produces in the behaviour of each model. Table 2 only shows the experiments involving the zooplankton maximal grazing rate ($R_m$), mortality of zooplankton (m), and the maximum value of the Chl : N ratio ($\theta_{max}^N$), as they were demonstrated to be the most significant parameters in terms of the sensitivity of the model according to the attributes calculated, with the rest of the experiments omitted for this discussion. The S2P3 v8.0 model is strongly influenced by NPZ parameters, with the zooplankton maximal grazing rate ($R_m$) (Stegert et al., 2007) and zooplankton mortality rate (m) having the largest effect in the magnitude of the spring phytoplankton bloom (Figures C1a, C4a) and in the total annual zooplankton biomass (Figures C1b, C4b). The effects of $R_m$ implies that lower values in the maximum ingestion rate of phytoplankton can produce earlier and larger spring phytoplankton blooms compared to the calibrated S2P3 v8.0 model. On the other hand, a higher value of $R_m$ shows a delayed spring phytoplankton bloom (Figure C2) compared to the CTD observations. Additionally, zooplankton mortality produced differences in the timing of the spring phytoplankton bloom, with delays of 30 days (year 2014) and 35 days (year 2015) when there is less zooplankton mortality, affecting the timing and magnitude of the spring zooplankton bloom, therefore, generating low values of surface chlorophyll-*a* during spring (Figure C5). It is well known that zooplankton are key players in the biogeochemical cycling of carbon and nutrients in marine ecosystems (Beaugrand and Kirby, 2010; Beaugrand et al., 2010), influencing the export of organic matter to the deep ocean (González et al., 2009; Juul-Pedersen et al., 2010). Additionally, grazing responses comprise the dominant losses for phytoplankton in the ocean (Banse, 1994), influencing plankton stocks and primary production (Franks et al., 1986).

Table 2 presents differences for the S2P3 v8.0 model calibrated for the CCS location and for selected sensitivity experiments in terms of: the timing of the spring phytoplankton bloom (days), defined as in Table B2, using a threshold for phytoplankton biomass (>1.5 mg Chl m$^{-3}$); the magnitude of the spring phytoplankton bloom (mg Chl m$^{-3}$); and the total annual zooplankton biomass (g DW m$^{-2}$). Representation of phytoplankton physiology had an important influence on the timing of the spring phytoplankton bloom (Table 2), with $\theta_{max}^N$ affecting the S2P3 v8.0 model the most in terms of this attribute of the model structure, showing less productive and delayed spring blooms when $\theta_{max}^N$ is lower (Figures C7a, C8b). Changes in the timing and magnitude of the spring zooplankton bloom coincides with the changes of the timing and magnitude of the phytoplankton blooms (Figure C7b). These changes in the plankton communities over the year due to different values of $\theta_{max}^N$, also have an

370 effect on the values of DIN (Figure C8c,f), with the largest differences shown during springtime at the surface (Figure C8c). These differences agree with the results found by Ayata et al. (2013), where it was demonstrated that taking into account photoacclimation and variable stoichiometry of phytoplankton growth in marine ecosystem models, produce qualitative and quantitative differences in phytoplankton dynamics. Moreover, these quota formulations in S2P3 v8.0 were compared to the available dataset of physiological observations (Figure C9). It is interesting to note that the sensitivity analysis of NPZ param-

375 eters produced differences in the physiological variables $P_{max}^{chl}$ and $E_k$, specially at the surface (Figures C3, C6), suggesting that the predator-prey interactions are indirectly influencing phytoplankton physiology, presumably through feedbacks between zooplankton and the nutrient cycling, which subsequently have an effect on phytoplankton physiology due to the dependency of nutrient quotas to the availability of inorganic nutrients. The current study thus demonstrates how a greater variety of data, spanning multiple trophic levels and incorporating information on physiological status as well as standing stocks, provides

additional constraints on model validation and hence constraints on parameterisation.

| Experiments | Years | Timing spring phytoplankton bloom (date) | Magnitude spring phytoplankton bloom (mg Chl m$^{-3}$) | Total annual zooplankton biomass (g DW m$^{-2}$) |
|---|---|---|---|---|
| Calibrated S2P3 v8.0 | 2014 | 3$^{rd}$ April | 137.7 | 6110 |
| | 2015 | 9$^{th}$ April | 172.6 | 6094 |
| $\theta_{max}^{N} \downarrow$ | 2014 | 19$^{th}$ May | 103.5 | 3055 |
| | 2015 | 21$^{st}$ May | 103.1 | 3514 |
| $\theta_{max}^{N} \uparrow$ | 2014 | 25$^{th}$ March | 110.2 | 6733 |
| | 2015 | 27$^{th}$ March | 211.5 | 6797 |
| $R_m \downarrow$ | 2014 | 29$^{th}$ March | 292.5 | 6840 |
| | 2015 | 3$^{rd}$ April | 359.2 | 6812 |
| $R_m \uparrow$ | 2014 | 14$^{th}$ April | 61.4 | 5283 |
| | 2015 | 25$^{th}$ April | 159.5 | 4992 |
| $m \downarrow$ | 2014 | 3$^{rd}$ May | 238 | 8362 |
| | 2015 | 14$^{th}$ May | 203.5 | 8369 |
| $m \uparrow$ | 2014 | 31$^{st}$ March | 104.4 | 4311 |
| | 2015 | 6$^{th}$ April | 134.5 | 4368 |

**Table 2.** List of the most sensitive experiments run for the S2P3 v8.0 model calibrated for the CCS location, including the year of observations, timing and magnitude of the spring phytoplankton bloom, and total annual zooplankton biomass values.

# 6 Comparison of overall model performance

The S2P3 v7.0 (Sharples et al., 2006; Marsh et al., 2015), S2P3-NPZ, S2P3-Photoacclim, and S2P3 v8.0 models were cali-
brated for the CCS location and further analysis was undertaken by running each model for a extended period (1965 - 2015),
to evaluate the statistics of productivity, partitioned between spring and summer. Table 3 shows that for the S2P3 v7.0 model,
on average, 69.2% of the annual phytoplankton production occurs during the spring phytoplankton bloom. On the other hand,
for the S2P3-NPZ model, on average, only 37.8% of the annual production occurs during spring months, showing that the
predator-prey relationship has a strong influence on the magnitude of the spring phytoplankton bloom every year. Additionally,
the S2P3-Photoacclim model shows a very strong spring phytoplankton bloom, corresponding to 90% of the total annual NPP.
Finally, for the S2P3 v8.0 model, only 67.4% of the annual production corresponds to the spring bloom period. It is clear that,
on average, the least productive model overall was S2P3 v8.0, followed by the S2P3-NPZ, S2P3 v7.0, and S2P3-Photoacclim.
This shows the impact and complexity that the predator-prey relationship has on the model dynamics, with the addition of
explicit zooplankton and their grazing activity as one of the main losses of phytoplankton (Franks et al., 1986). On the other
hand, the S2P3 v7.0 model has, on average, more total annual NPP than the S2P3-NPZ model, suggesting that the influence
of a constant grazing rate is not as strong in comparison to the one provided by the zooplankton grazing (NPZ framework),
because the predator-prey relationship can not be entirely represented in the S2P3 v7.0 model.

| Model | Characteristic (Units) | Mean | Maximum | Minimum | STD |
|---|---|---|---|---|---|
| **S2P3 v7.0** | Total spring NPP ($g\,C\,m^{-2}\,yr^{-1}$) | 39.6 | 54.7 | 33.5 | 5.8 |
| | Total summer NPP ($g\,C\,m^{-2}\,yr^{-1}$) | 17.2 | 22.9 | 56.9 | 5.1 |
| | Total annual NPP ($g\,C\,m^{-2}\,yr^{-1}$) | 57.1 | 61.1 | 53.1 | 1.8 |
| **S2P3-NPZ** | Total spring NPP ($g\,C\,m^{-2}\,yr^{-1}$) | 21.1 | 49.3 | 7.2 | 9.8 |
| | Total summer NPP ($g\,C\,m^{-2}\,yr^{-1}$) | 28.2 | 39.7 | 6.8 | 7.7 |
| | Total annual NPP ($g\,C\,m^{-2}\,yr^{-1}$) | 55.7 | 60.0 | 52.7 | 1.5 |
| **S2P3-Photoacclim** | Total spring NPP ($g\,C\,m^{-2}\,yr^{-1}$) | 35.6 | 40.7 | 25.7 | 4.8 |
| | Total summer NPP ($g\,C\,m^{-2}\,yr^{-1}$) | 3.8 | 11.4 | 0.6 | 4.2 |
| | Total annual NPP ($g\,C\,m^{-2}\,yr^{-1}$) | 39.4 | 42.4 | 37.0 | 1.7 |
| **S2P3 v8.0** | Total spring NPP ($g\,C\,m^{-2}\,yr^{-1}$) | 25.5 | 39.5 | 14.5 | 3.7 |
| | Total summer NPP ($g\,C\,m^{-2}\,yr^{-1}$) | 10.3 | 12.9 | 6.8 | 1.1 |
| | Total annual NPP ($g\,C\,m^{-2}\,yr^{-1}$) | 37.8 | 47.8 | 33.4 | 2.2 |

**Table 3.** Comparison between the S2P3 v7.0, S2P3-NPZ, S2P3-Photoacclim, and S2P3 v8.0 models calibrated for the CCS location, in terms
of the total spring NPP, total summer NPP, and total annual NPP calculated from 1965 to 2015, including the mean, maximum, minimum,
and STD values.

# 7 Conclusions

This study demonstrates that the combination of an NPZ framework, photo-acclimation, and flexible stoichiometry of phytoplankton in one model produces a better representation of the ecosystem based on the comparison to observations. This combined framework offers an improvement to the S2P3 v7.0 model, for application to the CCS location and more broadly within shelf sea systems. The model validation, using both zooplankton biomass and physiological rates of phytoplankton observations are rarely found in the literature, providing a novel contribution to the marine biogeochemistry modelling field of shelf seas.The development of the S2P3 v8.0 model provides a better fit to observations in comparison to the S2P3 v7.0, S2P3-NPZ, and S2P3-Photoacclim models. Improved confidence in the S2P3 v8.0 model thus suggest improved insights in studies about the effects of physical forcing through tides (Sharples, 2008), intra and inter-annual variations in meterology (Sharples et al., 2006) and other drivers on PP, and phytoplankton dynamics would be possible.

Appropriate parameterisations to represent shelf seas is a subject that should be further supported by fieldwork campaigns and future work should aim to include additional datasets with longer time-series (Friedrichs et al., 2007; Ward et al., 2010). For this study, constraining the seasonal cycle of the phytoplankton physiology is not possible due to the lack of physiological observations during other periods of the year, furthermore, phytoplankton and zooplankton biomass datasets only include the years 2014 and 2015 but longer time-series would help to improve the model calibration. Many model parameters quantities are poorly constrained observationally mainly due to the fact that model state variables are highly integrated pools, which are affected by biotic and abiotic factors in the environment, making them difficult to be determined by *in situ* measurements (Fennel et al., 2001).

The tuning and sensitivity analysis performed in this work allows a better understanding of the ecosystem dynamics represented in the model and how it is influenced by each parameter. By considering the timing and magnitude of the spring phytoplankton bloom, the annual zooplankton biomass, and summer phytoplankton photosynthetic physiology as features to be calculated, a quantitative and clearer comparison between each model could be developed. Finally, the model calibration will never be in perfect agreement with all the observations, particularly in this case, where only one type of phytoplankton and zooplankton were considered. Thus responses must represent some typical or average dynamics and cannot represent any effects of competition between types. Additionally no stages of zooplankton growth were taken into account, which might affect the predator-prey interactions (Wroblewski, 1982; Fennel, 2001). Despite such potential limitations, the S2P3 v8.0 model is able to reasonably represent the integrated behaviour of the mixture of species that inhabit the NW European Shelf Sea.

*Code and data availability.* S2P3 v8.0

The current version of model is available from https://doi.org/10.5281/zenodo.3600467 under the Creative Commons Attri-
bution 4.0 International license. The exact version of the model used to produce the results used in this paper is archived on
Zenodo (https://zenodo.org), as are input data and scripts to run the model and produce the plots for all the simulations pre-
sented in this paper.

Unzipped and uncompressed, the directory /s2p3v8.0 contains several sub-directories:

- /main contains the source code, s2p3_v8.f90, which is compiled "stand-alone", and executed using accompanying
  scripts.

- /domain contains, location (latitude and longitude), tidal components, and bathymetry (total depth) for the Central Celtic
  Sea in the western English Channel (s12_m2_s2_n2_h_tim.dat).

- /met contains 2000 - 2015 meteorological forcing ($Falmouth\_met\_i143\_j17\_2000-2015$ ASCII file).

- /output contains example output data from experiments that show the calibrated version of each model (S2P3v8, S2P3-
  Photoacclim, and S2P3-NPZ).

- /plotting contains MATLAB scripts for plotting time-series data from experiments S2P3v8, S2P3-Photoacclim, and
  S2P3-NPZ. This folder also include all the dataset from observations used to validate each model.

The ancillary files needed for simulations are available on request from the author (e-mail aab1g15@soton.ac.uk).

*Acknowledgements.* This work was funded by the CONICYT PFCHA/DOCTORADO BECAS CHILE/2015 - 72160249.

*Author contributions.* Angela A. Bahamondes Dominguez developed the model code, performed the simulations, and prepared the manuscript
with contributions from all co-authors.

*Competing interests.* The authors declare that they have no conflict of interest.

## Appendix A: Model variables and parameters

| Variables | Definition | Units |
|---|---|---|
| $I_{PAR}$ | Photosynthetically available radiation | $\mathrm{W\,m^{-2}}$ |
| $K_z$ | Depth-dependant vertical eddy diffusivity | $\mathrm{m^2\,s^{-1}}$ |
| $N$ | External DIN concentration | $\mathrm{mmol\,N\,m^{-3}}$ |
| $Phyto_C$ | Phytoplankton biomass in carbon currency | $\mathrm{mg\,C\,m^{-3}}$ |
| $Phyto_{chl}$ | Phytoplankton biomass in chlorophyll currency | $\mathrm{mg\,Chl\,m^{-3}}$ |
| $Phyto_N$ | Phytoplankton biomass in nitrogen currency | $\mathrm{mmol\,N\,m^{-3}}$ |
| $Z$ | Zooplankton biomass in nitrogen currency | $\mathrm{mmol\,N\,m^{-3}}$ |
| $I$ | Ingestion rate of phytoplankton | $\mathrm{d^{-1}}$ |
| $P_m$ | Carbon-specific rate of photosynthesis | $\mathrm{d^{-1}}$ |
| $u$ | Phytoplankton carbon-specific nitrate uptake rate | $\mathrm{mmol\,N\,(mg\,C)^{-1}\,d^{-1}}$ |
| $Q_P$ | Cellular nutrient quota (N:C) | $\mathrm{mmol\,N\,(mg\,C)^{-1}}$ |
| $Q$ | Internal nitrogen : phytoplankton chlorophyll ratio | $\mathrm{mmol\,N\,(mg\,Chl)^{-1}}$ |
| $z$ | Vertical coordinate (positive upwards) | $\mathrm{m}$ |
| $\theta$ | Chlorophyll : phytoplankton carbon ratio | $\mathrm{mg\,Chl\,(mg\,C)^{-1}}$ |
| $\mu$ | Carbon-specific rate of photosynthesis | $\mathrm{d^{-1}}$ |
| $\rho_{chl}$ | Chlorophyll synthesis regulation term | $\mathrm{mg\,Chl\,(mmol\,N)^{-1}}$ |

**Table A1.** List of all variables for the biological part of the S2P3 v8.0 model.

| Parameters | Definition | Units | Value |
|---|---|---|---|
| $k_n$ | Half-saturation constant for nitrate uptake | mmol N m$^{-3}$ | 0.014 |
| $P_{max}^C$ | Maximum value of the carbon-specific rate of photosynthesis | d$^{-1}$ | 3.5 |
| $Q_m$ | Maximum value of the cellular nutrient quota | mmol N (mg C)$^{-1}$ | 0.032 |
| $Q_{min}$ | Minimum value of the cellular nutrient quota | mmol N (mg C)$^{-1}$ | 0.0028 |
| $R_C$ | Respiration rate constant | d$^{-1}$ | 0.02 |
| $R_{chl}$ | Chlorophyll degradation rate constant | d$^{-1}$ | 0.02 |
| $R_n$ | Nitrate remineralisation rate constant | d$^{-1}$ | 0.02 |
| $u_m$ | Maximum value of the phytoplankton carbon-specific nitrate uptake rate | mmol N (mg C)$^{-1}$ d$^{-1}$ | 0.004 |
| $\alpha^{chl}$ | Chlorophyll-specific initial slope of the photosynthesis-light curve | mg C (mg Chl)$^{-1}$ d$^{-1}$ (W m$^{-2}$)$^{-1}$ | 1.99x10$^{-6}$ |
| $\zeta$ | Cost of biosynthesis | mg C (mmol N)$^{-1}$ | 0 |
| $\theta_{max}^N$ | Maximum value of the chlorophyll : phytoplankton nitrogen ratio | mg Chl (mmol N)$^{-1}$ | 2.1 |
| $T_{func}$ | Temperature-response function | dimensionless | 1.0 |
| m | Loss rate of zooplankton due to predation and physiological death | d$^{-1}$ | 0.02 |
| $R_m$ | Zooplankton maximal grazing rate | d$^{-1}$ | 3.5 |
| $\gamma_1$ | Grazing inefficiency or 'messy feeding' (0.0-1.0), returns a fraction of grazed material back into the DIN pool | dimensionless | 0.1 |
| $\gamma_2$ | Fraction of dead zooplankton (0.0-1.0) that goes into the sediments | dimensionless | 0.4 |
| $\lambda$ | Rate at which saturation is achieved with increasing food levels | (mmol N m$^{-3}$)$^{-1}$ | 0.014 |

**Table A2.** List of parameters, with their respective definitions, units, and initialised values for the biological part of the S2P3 v8.0 model.

**Appendix B:  Model calibration**

| Cruise name | Date | Latitude (°N) | Longitude (°W) | Depth (m) |
|---|---|---|---|---|
| DY029 | 03/04/2015 | 49.38 | 8.59 | 147 |
| DY029 | 04/04/2015 | 49.38 | 8.59 | 146 |
| DY029 | 05/04/2015 | 49.38 | 8.59 | 146 |
| DY029 | 06/04/2015 | 49.4 | 8.58 | 147 |
| DY029 | 11/04/2015 | 49.39 | 8.58 | 145 |
| DY029 | 15/04/2015 | 49.4 | 8.59 | 147 |
| **DY029** | **20/04/2015** | **49.4** | **8.6** | **147** |
| DY029 | 21/04/2015 | 49.4 | 8.62 | 148 |
| DY029 | 25/04/2015 | 49.4 | 8.59 | 148 |
| DY029 | 26/04/2015 | 49.4 | 8.58 | 146 |
| DY029 | 28/04/2015 | 49.4 | 8.58 | 146 |
| DY033 | 13/07/2015 | 49.43 | 8.59 | 144 |
| DY033 | 14/07/2015 | 49.42 | 8.54 | 144 |
| DY033 | 15/07/2015 | 49.37 | 8.61 | 145 |
| **DY033** | **24/07/2015** | **49.36** | **8.62** | **145** |
| DY033 | 25/07/2015 | 49.41 | 8.59 | 148 |
| DY033 | 29/07/2015 | 49.42 | 8.57 | 147 |
| DY033 | 30/07/2015 | 49.4 | 8.57 | 148 |
| DY033 | 01/08/2015 | 49.38 | 8.58 | 146 |

**Table B1.** List of relevant CTD casts for the CCS location from DY029 and DY033 cruises considering the date, location (latitude and longitude), and depth. CTD casts in red are the ones chosen in this work to validate each model during spring and summer.

| Years | Buoy | S2P3-NPZ | S2P3-Photoacclim | S2P3 v8.0 |
|---|---|---|---|---|
| 2014 | 5[th] April | 14[th] April | 25[th] April | 3[rd] April |
| 2015 | 16[th] March | 24[th] April | 6[th] May | 9[th] April |

**Table B2.** Quantitative comparison of the timing of the spring phytoplankton bloom between the buoy observations, S2P3-NPZ, S2P3-Photoacclim, and S2P3 v8.0 models.

## Appendix C: Sensitivity studies

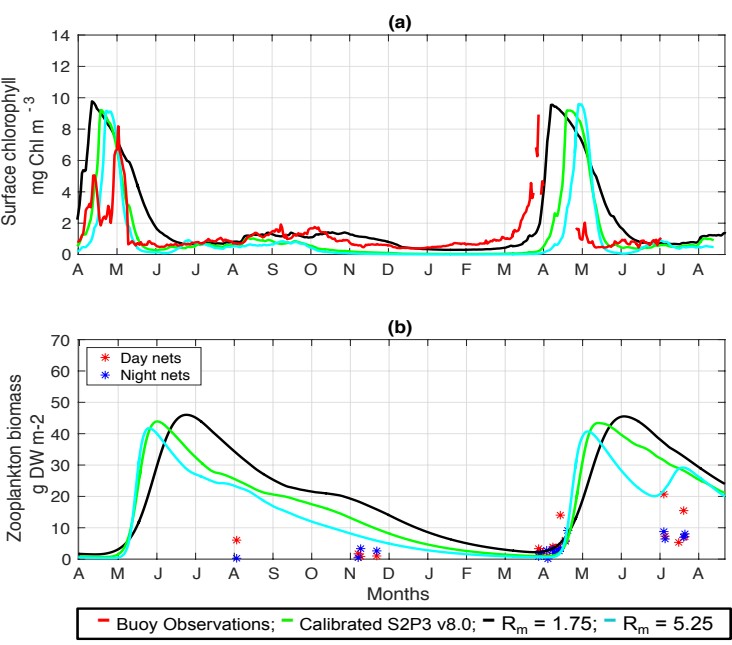

**Figure C1.** (a) SSB observations of surface chlorophyll-*a* (red line), along with the modelled surface chlorophyll-*a* for the calibrated S2P3 v8.0 model (green line), and experiments $R_m = 1.75$ (black line), and $R_m = 5.25$ (cyan line). (b) Observations of zooplankton biomass presented as discrete points for day nets (red dots) and night nets (blue dots) taken during the cruises DY026, DY018, DY029, and DY033; modelled zooplankton biomass from the calibrated S2P3 v8.0 model (green line) and experiments $R_m = 1.75$ (black line), and $R_m = 5.25$ (cyan line).

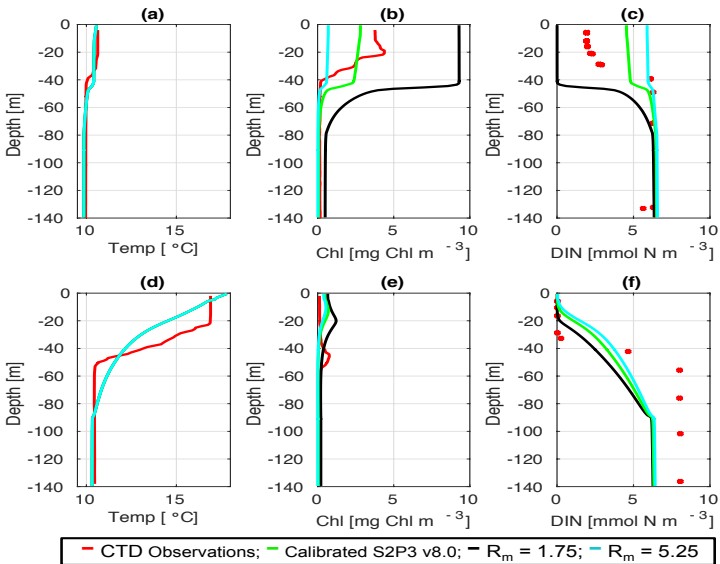

**Figure C2.** CTD observations from the SSB programme (red line) including data for: springtime (20/04/2015) (a) temperature, (b) chlorophyll-*a*, and (c) DIN (red dots); for summertime (24/07/2015) for (d) temperature, (e) chlorophyll-*a*, and (f) DIN (red dots) along the calibrated S2P3 v8.0 model (green line), experiments: $R_m = 1.75$ (black line), and $R_m = 5.25$ (cyan line).

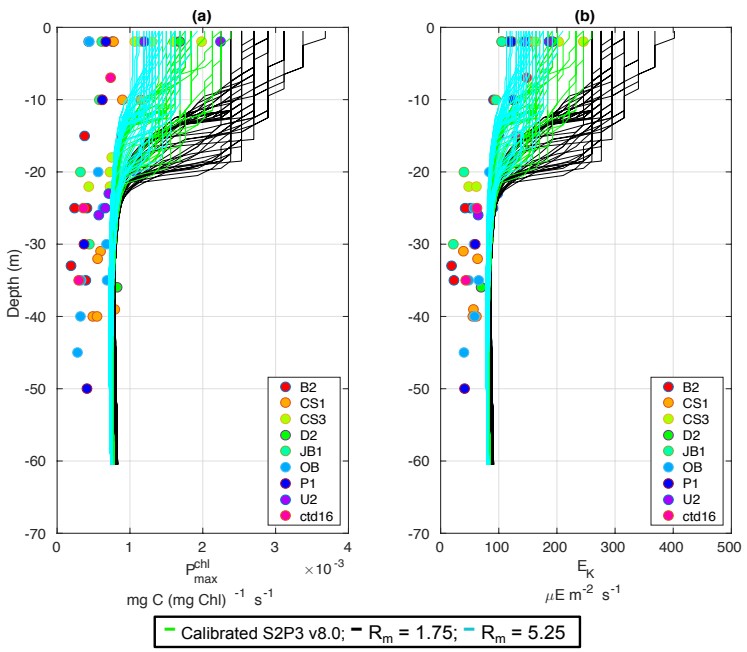

**Figure C3.** Observations from the cruises CD173 and JR98 in different locations of the Celtic Sea, including the calibrated S2P3 v8.0 model (green lines), experiments $R_m = 1.75$ (black lines), and $R_m = 5.25$ (cyan lines) for: (a) chlorophyll-*a* specific maximum light-saturated photosynthesis rate ($P_{max}^{Chl}$) and (b) light saturation parameter ($E_k$).

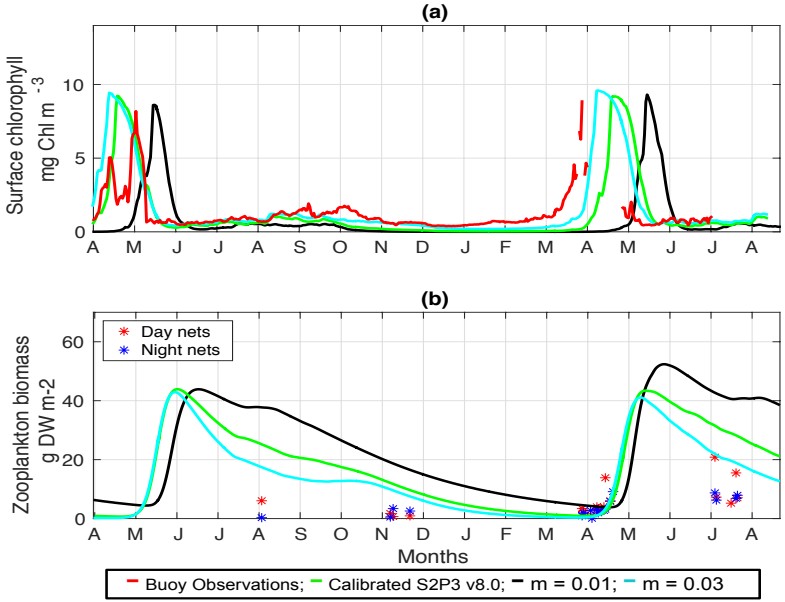

**Figure C4.** (a) SSB observations of surface chlorophyll-*a* (red line), along with the modelled surface chlorophyll-*a* for the calibrated S2P3 v8.0 model (green line), and experiments m = 0.01 (black line), and m = 0.03 (cyan line). (b) Observations of zooplankton biomass presented as discrete points for day nets (red dots) and night nets (blue dots) taken during the cruises DY026, DY018, DY029, and DY033; modelled zooplankton biomass from the calibrated S2P3 v8.0 model (green line) and experiments m = 0.01 (black line), and m = 0.03 (cyan line).

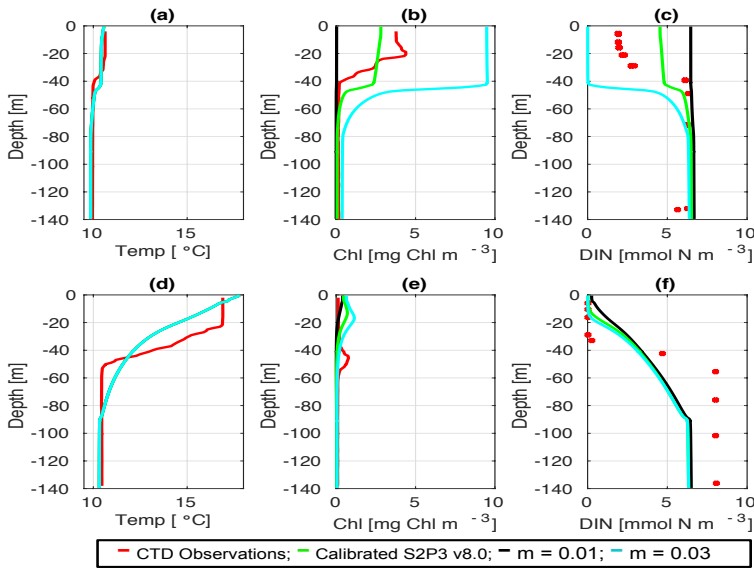

**Figure C5.** CTD observations from the SSB programme (red line) including data for: springtime (20/04/2015) (a) temperature, (b) chlorophyll-*a*, and (c) DIN (red dots); for summertime (24/07/2015) for (d) temperature, (e) chlorophyll-*a*, and (f) DIN (red dots) along the calibrated S2P3 v8.0 model (green line), experiments: $m = 0.01$ (black line), and $m = 0.03$ (cyan line).

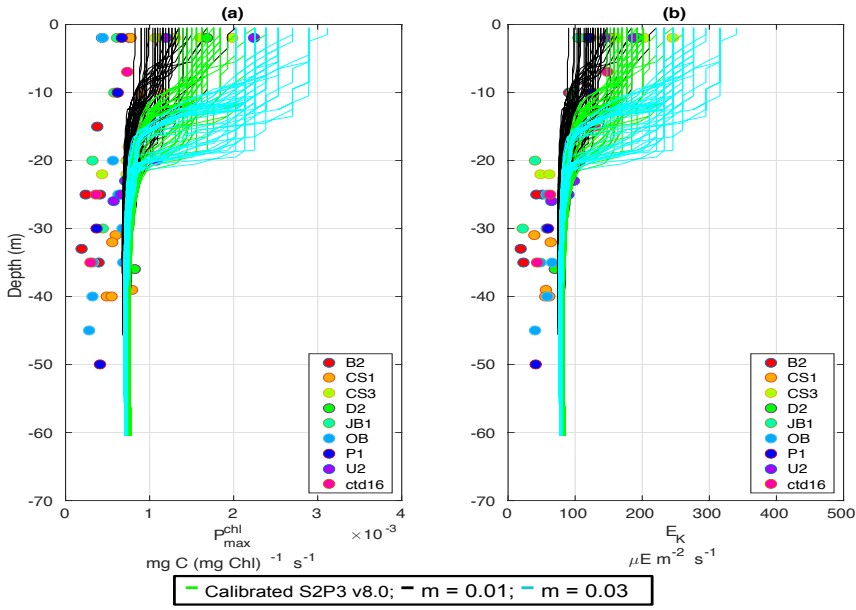

**Figure C6.** Observations from the cruises CD173 and JR98 in different locations of the Celtic Sea, including the calibrated S2P3 v8.0 model (green lines), experiments $m = 0.01$ (black lines), and $m = 0.03$ (cyan lines) for: (a) chlorophyll-*a* specific maximum light-saturated photosynthesis rate ($P_{max}^{Chl}$) and (b) light saturation parameter ($E_k$).

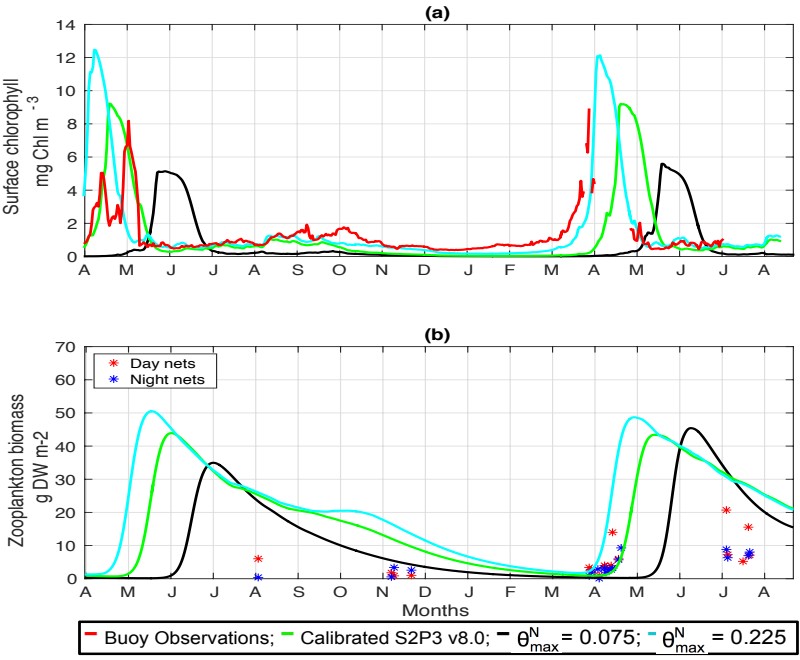

**Figure C7.** (a) SSB observations of surface chlorophyll-*a* (red line), along with the modelled surface chlorophyll-*a* for the calibrated S2P3 v8.0 model (green line), and experiments $\theta^N_{\mathrm{max}} = 0.075$ (black line), and $\theta^N_{\mathrm{max}} = 0.225$ (cyan line). (b) Observations of zooplankton biomass presented as discrete points for day nets (red dots) and night nets (blue dots) taken during the cruises DY026, DY018, DY029, and DY033; modelled zooplankton biomass from the calibrated S2P3 v8.0 model (green line) and experiments $\theta^N_{\mathrm{max}} = 0.075$ (black line), and $\theta^N_{\mathrm{max}} = 0.225$ (cyan line).

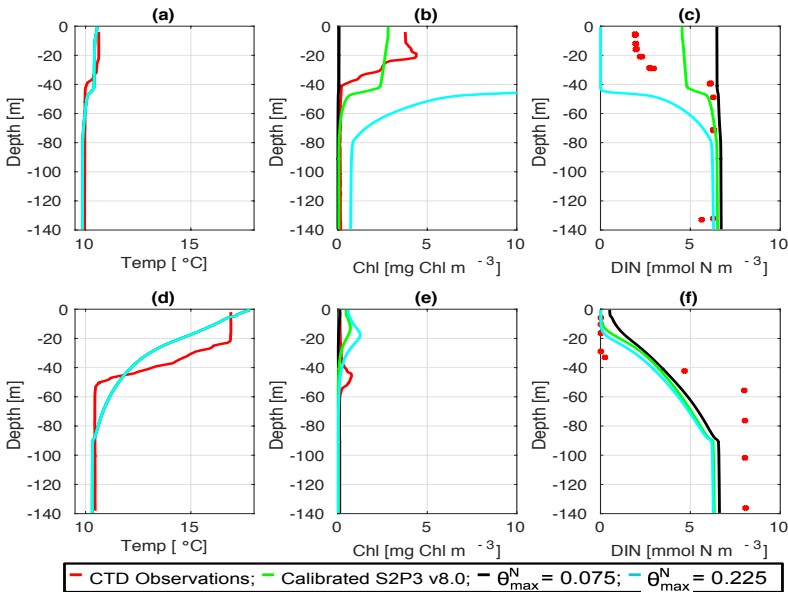

**Figure C8.** CTD observations from the SSB programme (red line) including data for: springtime (20/04/2015) (a) temperature, (b) chlorophyll-*a*, and (c) DIN (red dots); for summertime (24/07/2015) for (d) temperature, (e) chlorophyll-*a*, and (f) DIN (red dots) along the calibrated S2P3 v8.0 model (green line), experiments $\theta_{max}^N = 0.075$ (black line), and $\theta_{max}^N = 0.225$ (cyan line).

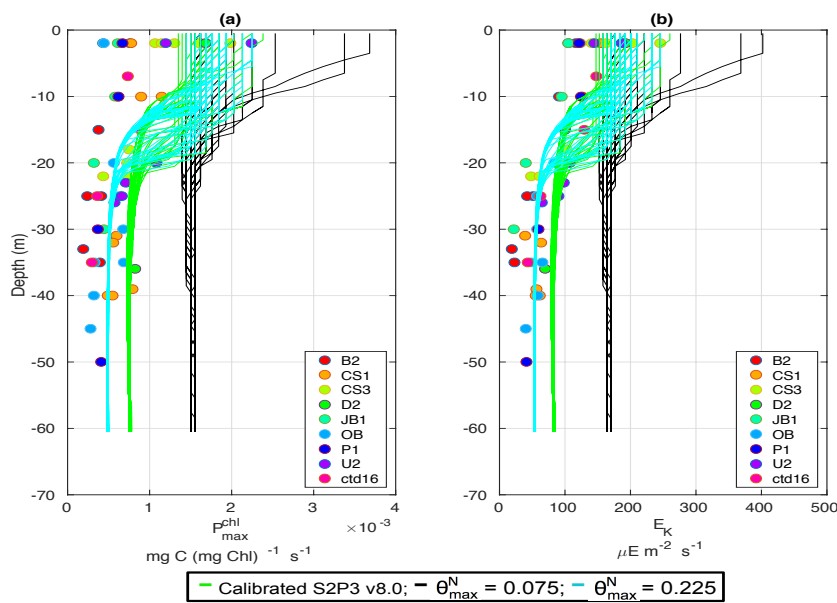

**Figure C9.** Observations from the cruises CD173 and JR98 in different locations of the Celtic Sea, including the calibrated S2P3 v8.0 model (green lines), experiments $\theta_{max}^N = 0.075$ (black lines), and $\theta_{max}^N = 0.225$ (cyan lines) for: (a) chlorophyll-*a* specific maximum light-saturated photosynthesis rate ($P_{max}^{Chl}$) and (b) light saturation parameter ($E_k$).

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
