# Peer review of "Constraining the response of phytoplankton to zooplankton grazing and photo-acclimation in a temperate shelf sea with a 1-D model - towards S2P3 v8.0"

_Geoscientific Model Development, 2019_

## Referee Comment (RC1) · Anonymous Referee #1 · 2 Mar 2020

General comments

The manuscript describes improvements made to a 1D nutrients-phytoplankton model in order to better describe the dynamics of the pelagic ecosystem in terms of phytoplankton growth and zooplankton biomass. Model simulations that included flexible stoichiometry and photo acclimation mechanisms for phytoplankton and variable grazing by zooplankton were compared with an ample set of observations collected in a temperate shelf sea. The integration of modelled and observed estimates of phytoplankton biomass and physiology, and zooplankton biomass, constitutes the main strength of the

work.

However, from my perspective, it is not clear under the current state of the manuscript what are the main findings of the work and why are they relevant. Model improvements are not innovative per se, and it is not clear how these improvements have contributed to provide insight into the dynamics of the ecosystem in such shelf sea.

Also, the model description is not very detailed and this makes difficult to understand which are exactly the model improvements that the authors are testing in this work. The description of sampling procedures is not complete, since relevant information about samples' collection and analysis techniques is missing.

Specific comments

Introduction

My main comment about the introduction is that is not clear from the text what is the point of the work, what does it add to the existing knowledge?

L50-54 This paragraph is difficult to understand. The authors say that they implemented flexible stoichiometry in the model and photoacclimation in the chlorophyll description, but they cite Droop which suggest that they have also included some functional relationship between growth rate and internal nutrient status or quotas. I think the authors should clarify which are exactly the modifications done in the model, only flexible stoichiometry or a full Droop's model.

L65 The authors mention here several models, but it is easy to get lost at this point. Perhaps a list of the models compared would be useful.

L69 "the effects of photo-acclimation and flexible stoichiometry" on what?

Methods

L107 Is that really internal nitrogen (m3 as phyto biovolume) or it is just phytoplankton nitrogen concentration in sea water? I believe is the latter, as in zooplankton (L108),

so the sentence should state nitrogen and not internal nitrogen.

L101-118 This model description is not very clear. Does S2P3-NPZ includes flexible stoichiometry? I guess it does since phytoplankton is described with Chla and N. If the ratio is fixed there is no need to compute both. The authors mentioned in the introduction that the approach used was a Droop model, where the growth rate has to depend on the internal quotas of nutrients. The way the authors describe the model it seems that S2P3-Photoacclim allows flexible stoichiometry but it is not clear how growth depends on quotas. It is also the growth dependent on quotas in S2P3-NPZ? I would suggest the authors to include the equations of the phytoplankton growth model they are using, to ensure clarity and reproducibility.

Figure 1. I imagine the idea of Figure 1 is to illustrate those equations without having to display them. Figure 1 is useful but not all symbols are explained. For instance, u, Z (zooplankton?) and QP are not described. Also, uptake of N in a) is u x PhytoChla but in b) to d) is u x PhytoC, I would say that a Q is necessary somewhere to convert units. And also in Figure 1, it is not clear how growth or uptake depend on quotas, if they do. Since one of the main goals of the manuscript is to account for flexible stoichiometry and hence simulate elemental quotas, maybe for improving clarity the authors could give clear names or symbols to each quota and refer to them throughout the manuscript.

L126. The description of the sampling performed during the SSB programme needs improvement: L129. Samples collected for what? Chlorophyll? L131. Please, indicate where this mooring is located. L132. I don't understand what info gives the term "stainless and titanium". L133. How these discrete samples were collected? with bottles? L134. The details about the CDT deployment in the other locations (not CCS) are missing.

L136. The description of the sampling of zooplankton biomass needs improvement: L138. How this zooplankton samples are collected? With nets? If so, please, specify

the type of net and the type of trawl. L140. How zooplankton biomass was measured?

L146. The description of the sampling for phytoplankton is also very vague: L148. Phytoplankton samples were collected with bottles? At "a number of stations" seems very vague.

Results

L177 The behaviour of S2P3-Photoacclim (blue line) seems very different to S2P3-NPZ, similar maybe in magnitude but not in timing.

L185 Zooplankton biomass increased 1 month later than phytoplankton in 2014 but definitely not in 2015.

L190 The expression "have not reached the SPB" is ambiguous. Maybe they haven't reached the peak of the bloom, but, at least for S2P3-NPZ, the accumulation of biomass has already started accordingly to Figure 3. It would be necessary to clarify to which event of the SPB the authors refer.

Figure 4. I miss the same figure for 2014, are CTD data not available for this year?

L203. The description of results regarding phytoplankton physiology seems a bit short and does not give much information about the agreement of the model and observations: L205 If it is a 1D model, are not the vertical gradients the only ones available? Figure 5. The vertical gradients are difficult to see for observations, especially for Pmax.

L218. To which location correspond Figures 6 and 7?

L236. In Figure 7 I don't feel it is possible to see which event follows which. Maybe a mean seasonal cycle with some metric for interannual variability would be more easy to interpret.

Figure 7 legend. Does this "forced with all meteorological components" refer to the whole model result? I imagine it does, but here it seems it refers to NPP only.

L244. Parameters listed in Table 1. The reader does not know where these parameters fit into the model. They are not in Figure 1 and no equation that includes them is shown in the manuscript.

Not all A Figures are described, are all necessary?

L276. The meaning of this sentence is not clear to me.

Table 2. It is not clear what this "timing of the SPB" means. Is it day of the onset, day of the peak? DW is dry-weight? If I am not wrong, it is not explained anywhere in the manuscript.

To which location/area correspond the metrics in Table 2 and Table 3?

Conclusions

L300. I am afraid I wouldn't call this work innovative. NPZ models that include grazing are commonly used. Flexible stoichiometry I would say is almost the norm when describing phytoplankton pools. And the description of photoacclimation through the parameterization proposed by (Geider et al. 1998) is virtually standard in biogeochemical models.

L305. This last sentence is difficult to understand. And also, I miss one sentence or paragraph that clearly states why this work is an advance in our knowledge.

Technical corrections

L18 SSB acronym not defined.

I would say "data" is plural (datum is the singular) and has been used as singular in several places (L20, L75, L82).

Sections 2 and 3 use mainly present tense, but there are several pasts in between that maybe can be reviewed (L76, L82, L103).

L203 "Moreover" and "also" in the same sentence sound redundant.

[Figure]

L250. I am not sure, but I feel it is more clear to refer to ingestion rate of zooplankton and not phytoplankton.

---

## Referee Comment (RC2) · Anonymous Referee #2 · 17 Mar 2020

Dear Sirs (English version):

The model presented by Angela Bahamondes Dominguez is interesting and it is related to the scope of the journal (Geoscientific Model Development). Overall, I find the model showing a considerable fit with the field data, and this suggest that could be useful to improve our understanding about how phytoplankton blooms take place in situ. For all these reasons, I think this article is within the standards of excellence of the journal.

The model presented (S2P3 v8.0) is a modification of a previous NPZ model (S2P3

v7.0) to which the photo-acclimatization of phytoplankton is introduced. This new model improves the previous model, as the results of the new model are better adjusted to in situ observations. However, the temperature is taken into account only in the respiration process, while it is not considered in photoacclimation and grazing, and both processes depend on it (Sarmento et al . 2010, Vázquez-Domínguez et al. 2013 and references therein). Furthermore, the model is slightly decoupled to the field data, as it happens in the second period (Fig. 3a) or the zooplankton biomass (Fib 3b), and besides it presents a mismatch with nitrogen (Figure 4c). This should be discussed.

Two additional questions: 1) it would be good to unify units in figure 6 since they are mixed (Chla, N, C), and this does not allow to estimate a transfer efficiency between trophic levels. If a scale with mmol C m-3 is added, the efficiency in carbon transfer can be estimated between phytoplankton-zooplankton, and if the same is done with N we would know the transfer between N-P-Z. This is important at the biogeochemical level; and, 2) similarly, perhaps two columns could be added in Table2, indicating the mg C m-2 of phytoplankton and zooplankton.

Finally, figure 7 (a) shows an inter-annual change in the intra-annual temperature variability, which may be due to temperature changes at the decadal level. All these changes seem to affect the biomass of phytoplankton, but they are not so apparent in the remaining variables. Perhaps, you have an explanation of these differences.

Yours sincerely

Sarmento et al. 2010. PTRS (b): Biol. Sci. 365:1549, 2137-2149 Vazquez-Dominguez et al. 2013. MEPS. 493: 43-56

Estimados Señores (Spanish version):

El modelo presentado por Angela Bahamondes Dominguez es interesante y se encuentra relacionado con el objetivo de la revista (Geoscientific Model Development). En general, encuentro que el modelo muestra un ajuste considerable con los datos de

campo. Por todo ello, creo que el artículo se encuentra dentro de los estándares de excelencia de la revista.

El modelo (S2P3 v8.0 ) es una modificación de un modelo NPZ anterior (S2P3 v7.0) al que se le introduce la foto-aclimatación del fitoplancton. Esto mejora el modelo anterior, dado que los resultados del nuevo modelo se ajustan mejor a las observaciones biológicas in situ. Sin embargo, desde mi punto de vista, subyace el problema de que la temperatura solamente se tiene en cuenta en la respiración, no se considera ni en la foto-aclimatación ni en la predación, siendo estos dos procesos dependientes de esta variable (Sarmento et al. 2012, Vázquez-Domínguez et al. 2013 y referencias citadas). Además, el modelo se encuentra desacoplado con los datos de campo en el segundo periodo (Fig. 3a) y en la biomasa de zoopankton respecto al modelo anterior. Esto convendría discutirlo en el manuscrito. Del mismo modo, el modelo presenta un desajuste con los datos de nitrógeno en superficie (figura 4C).

Dos cosas adicionales: 1) en la figura 6 estaría bien unificar unidades, dado que están mezcladas (Chla, N, C) y eso no permite estimar una eficiencia de transferencia entre niveles tróficos. Si se añadiese una escala con los mmol C m-3 podríamos ver que eficiencia en la transferencia de carbono nos está indicando el modelo entre fitoplancton-zooplancton, o si lo hacemos con N sabríamos la transferencia entre N-P-Z. Esto es importante a nivel biogeoquímico; 2) de la misma manera ocurre en la Tabla2, tal vez se podrían añadir dos columnas indicando los mg C m-2 de fitoplancton y zooplancton.

Finalmente, en la figura 7(a) se ve un cambio inter-anual en la variabilidad de temperatura intra-anual. Esto se puede deber a cambios de temperatura a nivel decadal. Además, estos cambios parece que afectan a la biomasa del fitoplancton, pero no son tan aparentes en el resto de variables. Tal vez valdría la pena discutir a que puede ser debida esa diferencia.

Atentamente

---

## Author Comment (AC1) · 20 May 2020

**Referee 1**
**General comments**

**Referee comment:** The manuscript describes improvements made to a 1D nutrients-phytoplankton model in order to better describe the dynamics of the pelagic ecosystem in terms of phytoplankton growth and zooplankton biomass. Model simulations that included flexible stoichiometry and photo acclimation mechanisms for phytoplankton and variable grazing by zooplankton were compared with an ample set of observations collected in a temperate shelf sea. The integration of modelled and observed estimates of phytoplankton biomass and physiology, and zooplankton biomass, constitutes the main strength of the work.

However, from my perspective, it is not clear under the current state of the manuscript what are the main findings of the work and why are they relevant. Model improvements are not innovative per se, and it is not clear how these improvements have contributed to provide insight into the dynamics of the ecosystem in such shelf sea.

Also, the model description is not very detailed and this makes difficult to understand which are exactly the model improvements that the authors are testing in this work. The description of sampling procedures is not complete, since relevant information about samples' collection and analysis techniques is missing.

**Response:** Thank you for your overall comment in this work, which provides a general overview of what was done and comments on the novelty of this paper. GMD is a scientific journal dedicated to the public discussion of the description, development, and evaluation of numerical models as described in the website. This paper was uploaded as a "Development and technical paper", for which the aim is to describe technical developments and model improvements. We think that this paper meets the requirements to be published under the standards of GMD and the manuscript type selected as the aim is to show the improvement in a published model (S2P3 v7.0) based on comparison to observations, which are demonstrated to have a better agreement with the dataset used for the CCS location. We demonstrate an improvement in assessing the dynamics of the selected region, increasing confidence in the results with the model (S2P3 v8.0). Furthermore, this paper also highlights the relative importance of certain processes (e.g. zooplankton grazing and photo-acclimation), which under the general perspective of shelf sea modeling are processes that can be better understood with the sensitive analysis done in this work and which should be taken into account for accuracy in the representation of the shelf sea ecosystem.

However, we understand that the initial manuscript might have had some confusing sections based on the comments of this reviewer and we will revise the manuscript to allow clearer understanding of the model development and specific relevance to the biogeochemistry of shelf seas.

**Changes in manuscript:** Please see our responses to following specific comments and the associated changes.

***Specific comments***

*Introduction*

**Referee comment:** My main comment about the introduction is that is not clear from the text what is the point of the work, what does it add to the existing knowledge?

**Response:** The point of the work is substantial development of the biological component of the model, as previously reported in Marsh et al. (2015). This is already clearly articulated in the Introduction (see L 26-42).

**Changes in manuscript:** None needed.

**Referee comment:** L50-54 This paragraph is difficult to understand. The authors say that they implemented flexible stoichiometry in the model and photoacclimation in the chlorophyll description, but they cite Droop which suggest that they have also included some functional relationship between growth rate and internal nutrient status or quotas. I think the authors should clarify which are exactly the modifications done in the model, only flexible stoichiometry or a full Droop's model.

**Response:** As stated in the indicated section, we implemented the Geider et al. 1998 model, which does indeed link growth to internal cellular nutrient status. We have clarified this in the text

**Changes in manuscript:** Moreover, changes in nutrient availability can further alter cellular chlorophyll and nitrogen quotas (Droop, 1983; Geider et al., 1998). The combined representation of these two processes in the physiological model of Geider et al. (1998) has been widely implemented in biogeochemical models (Moore et al. 2002). We term this new version of S2P3 v7.0 as S2P3-Photoacclim, which relates phytoplankton growth rates to cell quota (Droop, 1983), through linking the light-, nutrient-, and temperature-dependencies of phytoplankton growth rate to varying ratios of N : C : Chl (Geider et al., 1998).

**Referee comment:** L65 The authors mention here several models, but it is easy to get lost at this point. Perhaps a list of the models compared would be useful.

**Response:** We agree that some context is helpful here.

**Changes in manuscript:** Each characteristic is now related to a model in this sentence to about confusion. Reference to Figure 1 also allows clarification.

**Referee comment:** L69 "the effects of photo-acclimation and flexible stoichiometry" on what?

**Response:** We refer to the effects of these processes in the CCS location and to the response of phytoplankton to changes in their environment.

**Changes in manuscript:** Further comments and details have been added (see L69-72).

*Methods*

**Referee comment:** L107 Is that really internal nitrogen (m3 as phyto biovolume) or it is just phytoplankton nitrogen concentration in sea water? I believe is the latter, as in zooplankton (L108), so the sentence should state nitrogen and not internal nitrogen.

**Response:** Yes, it corresponds to phytoplankton nitrogen concentration.

**Changes in manuscript:** We have amended descriptions as 'internal nitrogen', rather than 'nitrogen'.

**Referee comment:** L101-118 This model description is not very clear. Does S2P3-NPZ includes flexible stoichiometry? I guess it does since phytoplankton is described with Chla and N. If the ratio is fixed there is no need to compute both. The authors mentioned in the introduction that the approach used was a Droop model, where the growth rate has to depend on the internal quotas of nutrients. The way the authors describe the model it seems that S2P3-Photoacclim allows flexible stoichiometry but it is not clear how growth depends on quotas. It is also the growth dependent on quotas in S2P3-NPZ? I would suggest the authors to include the equations of the phytoplankton growth model they are using, to ensure clarity and reproducibility.

**Response:** S2P3-NPZ includes a variable Chl:N as in S2P3 v7.0. For S2P3-Photoacclim the Chl:N:C varies, according to Geider et al. 1998 and in the source code of the model provided in Code and Data availability section. The choice of not including equations here and providing details of the equations with Figure 1

was decided based on the already published equations from different manuscripts (Geider et al., 1998, Sharples et al. 2006, Marsh et al., 2015). But we agree that further information about the S2P3 v8.0 should be provided with the equations.

**Changes in manuscript:** We will provide additional model details in Supplementary material (see Appendix A).

**Referee comment:** Figure 1. I imagine the idea of Figure 1 is to illustrate those equations without having to display them. Figure 1 is useful but not all symbols are explained. For instance, u, Z (zooplankton?) and QP are not described. Also, uptake of N in a) is u x PhytoChla but in b) to d) is u x PhytoC, I would say that a Q is necessary somewhere to convert units. And also in Figure 1, it is not clear how growth or uptake depend on quotas, if they do. Since one of the main goals of the manuscript is to account for flexible stoichiometry and hence simulate elemental quotas, maybe for improving clarity the authors could give clear names or symbols to each quota and refer to them throughout the manuscript.

**Response:** It is difficult to describe all those variables and parameters with one figure.

**Changes in manuscript:** As noted above, we will provide additional model details in Supplementary Material.

**Referee comment:** L126. The description of the sampling performed during the SSB programme needs improvement: L129. Samples collected for what? Chlorophyll? L131. Please, indicate where this mooring is located. L132. I don't understand what info gives the term "stainless and titanium". L133. How these discrete samples were collected? with bottles? L134. The details about the CDT deployment in the other locations (not CCS) are missing.

**Response:** We agree that details of observations are appropriate.

**Changes in manuscript:** For this set of questions about section 4.1, more details are provided for the SSB programme in terms of surface chlorophyll, indicating where the mooring was located (lat, lon), how and where the discrete samples were collected (Niskin bottles), and further details of other CTD deployments in the Celtic Sea are provided in Table A1. The term of 'stainless and titanium' has been removed as an unnecessary detail.

**Referee comment:** L136. The description of the sampling of zooplankton biomass needs improvement: L138. How this zooplankton samples are collected? With nets? If so, please, specify the type of net and the type of trawl. L140. How

**Response:** Again, we agree that further measurement details here are informative, although we note that a fuller description is provided in the published paper describing this data.

**Changes in manuscript:** A more detailed description of the zooplankton biomass sampling in section 4.2 has been added. This further detailed description includes information about how the zooplankton was collected (using nets of different sizes). Also, it is explained how zooplankton biomass was measured (using a FlowCam and ZooScan).

**Referee comment:** L146. The description of the sampling for phytoplankton is also very vague: L148. Phytoplankton samples were collected with bottles? At "a number of stations" seems very vague.

**Response:** Again, we agree that further measurement details here are informative.

**Changes in manuscript:** Further description of the physiological observations is detailed in section 4.3 now. Phytoplankton were collected at different stations in the Celtic Sea and at different depths using Niskin bottles. Please see these added details in first paragraph of section 4.3.

*Results*

**Referee comment:** L177 The behaviour of S2P3-Photoacclim (blue line) seems very different to S2P3- NPZ, similar maybe in magnitude but not in timing.

**Response:** Given further consideration about the physiological observations used, it was decided that the stations CS2 and N1 should not be used for the validation of the S2P3-Photoacclim and S2P3 v8.0 models because one of the important assumptions of these models is that they do not consider advective fluxes and these stations are very close to the shelf edge where horizontal advection is relevant. This allowed better constraint of S2P3-Photoacclim and, therefore, it presents an improved representation of the ecosystem (timing and magnitude of the spring bloom are more similar to the buoy observations, Figures 3a, 4).

**Changes in manuscript:** For new values constrained in this model see Table 1. Table 3 values of S2P3-Photoacclim are also amended.

**Referee comment:** L185 Zooplankton biomass increased 1 month later than phytoplankton in 2014 but definitely not in 2015.

**Response:** We agree that this is worthy of note.

**Changes in manuscript:** Section 4.4, L207, sentence rephrased specifying that the 1 month difference occurs during 2014, while in 2015 there is not.

**Referee comment:** L190 The expression "have not reached the SPB" is ambiguous. Maybe they haven't reached the peak of the bloom, but, at least for S2P3-NPZ, the accumulation of biomass has already started accordingly to Figure 3. It would be necessary to clarify to which event of the SPB the authors refer.

**Response:** The timing of the spring phytoplankton bloom corresponds to a specific threshold.

**Changes in manuscript:** We now clarify and detail further as the term 'reaching the spring phytoplankton bloom' according to different definitions in the literature. Based on this definition, dates for the spring phytoplankton bloom being reached by each model are now also specified. The corresponding sentence is rephrased (L215).

**Referee comment:** Figure 4. I miss the same figure for 2014, are CTD data not available for this year?

**Response:** CTD observations were available for the CCS location in 2014 only during November, otherwise the year 2014 would have been included to compare CTD casts during spring and summer as in the year 2015.

**Changes in manuscript:** None needed.

**Referee comment:** L203. The description of results regarding phytoplankton physiology seems a bit short and does not give much information about the agreement of the model and observations: L205 If it is a 1D model, are not the vertical gradients the only ones available? Figure 5. The vertical gradients are difficult to see for observations, especially for Pmax.

**Response:** We agree that further details should be provided in the description of the results that compare phytoplankton physiology between the model and observations.

**Changes in manuscript:** Further description of the extent to which the modelled values and vertical gradients in Figure 5 match the data will be added.

**Referee comment:** L218. To which location correspond Figures 6 and 7?

**Response:** All this paper is based at the CCS location, with all models calibrated

for that specific region.

**Changes in manuscript:** This is specified again in this sentence, which is also rephrased (L244).

**Referee comment:** L236. In Figure 7 I don't feel it is possible to see which event follows which. Maybe a mean seasonal cycle with some metric for interannual variability would be more easy to interpret.

**Response:** We agree to include details of the seasonal cycle and associated variability.

**Changes in manuscript:** Figure 7 has been modified to show the seasonal cycle of each variable in the S2P3 v8.0 model (black lines) and with the red lines representing the 95% quantiles (i.e. 95% of the data lie between these lines) of each variable over the 51 years of simulation to represent the inter-annual variability present in the dynamics of the ecosystem.

**Referee comment:** Figure 7 legend. Does this "forced with all meteorological components" refer to the whole model result? I imagine it does, but here it seems it refers to NPP only.

**Response:** We agree that clarification is helpful.

**Changes in manuscript:** Legend of Figure 7 rephrased and more details added to avoid this confusion.

**Referee comment:** L244. Parameters listed in Table 1. The reader does not know where these parameters fit into the model. They are not in Figure 1 and no equation that includes them is shown in the manuscript.

**Response**: See response to earlier comments.

**Changes in manuscript:** As noted above, we will provide additional model details in Supplementary Material (see Appendix A).

**Referee comment:** Not all A Figures are described, are all necessary?

**Response**: We agree that further details were needed in the description of those figures. They were all described but not explicitly referred to which figure it corresponded.

**Changes in manuscript:** Reference to each figure has been added and further details to show the relevance of including each of them in the Appendix (see

Section 5).

**Referee comment:** L276. The meaning of this sentence is not clear to me.

**Response:** Noted. This sentence has been re-written in a clearer way

**Changes in manuscript:** See L322 for changes in this sentence.

**Referee comment:** Table 2. It is not clear what this "timing of the SPB" means. Is it day of the onset, day of the peak? DW is dry-weight? If I am not wrong, it is not explained anywhere in the manuscript.

**Response:** Timing of the spring phytoplankton bloom is referred as the quantitative definition for the start of the spring phytoplankton bloom.

**Changes in manuscript:** This is now described in section 4.4, L217. In the description of Table 2, it has been specified that corresponds to such definition. For the DW units of zooplankton, they are now explained in section 4.2, for zooplankton biomass dataset, L165.

**Referee comment:** To which location/area correspond the metrics in Table 2 and Table 3?

**Response:** For the CCS location.

**Changes in manuscript:** Specified now in the legend of Tables 2, 3, and in the description of each table (L298, LL314).

*Conclusions*

**Referee comment:** L300. I am afraid I wouldn't call this work innovative. NPZ models that include grazing are commonly used. Flexible stoichiometry I would say is almost the norm when describing phytoplankton pools. And the description of photoacclimation through the parameterization proposed by (Geider et al. 1998) is virtually standard in biogeochemical models. L305. This last sentence is difficult to understand. And also, I miss one sentence or paragraph that clearly states why this work is an advance in our knowledge.

**Response:** As we explained in response to general comments, the advance of this work is substantial development of a model originally described in GMD.

**Changes in manuscript:** None needed.

*Technical corrections*

**Referee comment:** L18 SSB acronym not defined.

**Change to manuscript:** Amended. Defined now in L18.

**Referee comment:** I would say "data" is plural (datum is the singular) and has been used as singular in several places (L20, L75, L82).

**Change to manuscript:** Noted. Lines where 'data' is used, it is changed as plural in their description. These include L20, L75, L82.

**Referee comment:** Sections 2 and 3 use mainly present tense, but there are several pasts in between that maybe can be reviewed (L76, L82, L103).

**Change to manuscript:** Noted and amended in sections 2 and 3.

**Referee comment:** L203 "Moreover" and "also" in the same sentence sound redundant.

**Change to manuscript:** Sentence amended. Moreover only included (L234).

**Referee comment:** L250. I am not sure, but I feel it is more clear to refer to ingestion rate of zooplankton and not phytoplankton.

**Response:** We understand why it is confusing thanks to your comment. To avoid further confusion, we will refer to Rm as: zooplankton maximal grazing rate.

**Change to manuscript:** Amended in L250, L253, Table 1.

---

## Author Comment (AC2) · 20 May 2020

**Referee 2**

Dear Sirs (English version):

**Referee comment:** The model presented by Angela Bahamondes Dominguez is interesting and it is related to the scope of the journal (Geoscientific Model Development). Overall, I find the model showing a considerable fit with the field data, and this suggest that could be useful to improve our understanding about how phytoplankton blooms take place in situ. For all these reasons, I think this article is within the standards of excellence of the journal.

**Response:** Thank you for your comment regarding the standards of this paper. We think that describing developments in a model at a technical level complies with the aims of Geoscientific Model Development.

**Referee comment:** The model presented (S2P3 v8.0) is a modification of a previous NPZ model (S2P3 v7.0) to which the photo-acclimatization of phytoplankton is introduced. This new model improves the previous model, as the results of the new model are better adjusted to in situ observations. However, the temperature is taken into account only in the respiration process, while it is not considered in photoacclimation and grazing, and both processes depend on it (Sarmento et al . 2010, Vázquez-Domínguez et al. 2013 and references therein). Furthermore, the model is slightly decoupled to the field data, as it happens in the second period (Fig. 3a) or the zooplankton biomass (Fib 3b), and besides it presents a mismatch with nitrogen (Figure 4c). This should be discussed.

**Response**: We appreciate the points raised by the reviewer. Firstly, we are aware of the effects of temperature on phytoplankton physiology and grazing. Within the developed model iteration we explicitly include the representation of temperature on phytoplankton physiology using the model of Geider et al. (1998). We acknowledge that representation of temperature effects on zooplankton grazing could also have been included. However, we note that there are always further processes which could be included and decided that the development step of the model described here was substantive enough to be worth a formal description within GMD. We note that there remains discrepancy between the model and the data, indicating further potential avenues for development.

**Changes to manuscript:** We will add the full equations of S2P3 v8.0 in Supplementary Material (see Appendix A). Furthermore, regarding the no-temperature dependencies of grazing and photo-acclimation in this model, a sentence is added regarding this assumption in L216. Finally, in L225-226 is explained about the differences in DIN for Figure 4c.

**Referee comment:** Two additional questions: 1) it would be good to unify units in figure 6 since they are mixed (Chla, N, C), and this does not allow to estimate a transfer efficiency between trophic levels. If a scale with mmol C m-3 is added, the efficiency in carbon transfer can be estimated between phytoplankton-zooplankton, and if the same is done with N we would know the transfer

between N-P-Z. This is important at the biogeochemical level; and, 2) similarly, perhaps two columns could be added in Table2, indicating the mg C m-2 of phytoplankton and zooplankton.

**Response:** We agree that showing the transfer efficiency between trophic levels should be added.

**Changes to manuscript:** Figure 7 is changed to more easily represent this transfer efficiency, by adding a subpanel of phytoplankton N. Therefore, the transfer efficiency between trophic levels can be seen in terms of mmol N m-3 between phytoplankton-zooplankton-nutrients.

**Referee comment:** Finally, figure 7 (a) shows an inter-annual change in the intra-annual temperature variability, which may be due to temperature changes at the decadal level. All these changes seem to affect the biomass of phytoplankton, but they are not so apparent in the remaining variables. Perhaps, you have an explanation of these differences.

**Response:** We agree that this variability can be more clearly presented.

**Changes in manuscript:** Figure 7 has been modified to give the reader better insights about the dynamics of the model. The new figure shows the annual seasonal cycle of each variable (black line) and with the red lines representing the 95% quantiles (i.e. 95% of the data lie between these lines) of each variable over the 51 years of simulation to show the inter-annual variability of each variable. It is more apparent in this figure that not only biomass of phytoplankton shows changes through each year, but also all the remaining variables.